


# eSCAPE: Regional to Global Scale Landscape Evolution Model v2.0

Tristan Salles[1]

[1]School of Geosciences, University of Sydney, Sydney, NSW, 2006, Australia

**Correspondence:** Tristan Salles (tristan.salles@sydney.edu.au)

**Abstract.** eSCAPE is a Python-based landscape evolution model that simulates over geological time (1) the dynamic of the landscape, (2) the transport of sediment from source to sink, and (3) continental and marine sedimentary basins formation under different climatic and tectonic conditions. eSCAPE is open-source, cross-platform, distributed under the GPLv3 license and available on GitHub (escape-model.github.io). Simulated processes rely on a simplified mathematical representation of

landscape processes - the stream power and creep laws - to compute Earth's surface evolution by rivers and hillslope transport. The main difference with previous models is in the underlying numerical formulation of the mathematical equations. The approach is based on a series of implicit iterative algorithms defined in matrix form to calculate both drainage area from multiple flow directions and erosion/deposition processes. eSCAPE relies on PETSc parallel library to solve these matrix systems. Along with the description of the algorithms, examples are provided and illustrate the model current capabilities and

limitations. eSCAPE is the first landscape evolution model able to simulate processes at global scale and is primarily designed to address problems on large unstructured grids (several millions of nodes).

## 1 Introduction

Since the 80s, many software have been designed to estimate long-term catchment dynamic, drainage evolution as well as

sedimentary basins formation in response to various mechanisms such as tectonic or climatic forcing (Braun and Sambridge, 1997; Coulthard et al., 2002; Davy and Lague, 2009; Simoes et al., 2010; Salles, 2016; Grieve et al., 2016b; Hobley et al., 2017). These models rely on a set of mathematical and physical expressions that simulate sediment erosion, transport and deposition and can reproduce the first order complexity of Earth's surface geomorphological evolution (Tucker and Hancock, 2010; Shobe et al., 2017).

In most of these models, climatic and tectonic conditions are imposed and often consist in rather simple forcing such as uniform spatial precipitation and vertical displacements (uplift or subsidence) far from reflecting the complexity of the natural system. In addition such approaches are unable to properly explore potential feedback mechanisms between each of the Earth components. In fact, only a handful of these models are able to account more completely for the dynamics of the lithosphere and mantle, the role of sedimentation and provide a more quantitative representation of climate relative to its interactions with





topography (such as orographic rain) (Beaumont et al., 1992; Salles et al., 2011; Thieulot et al., 2014; Yang et al., 2015; Salles et al., 2017; Beucher et al., 2019). When made possible, it is often realised through the coupling of specialised numerical models involving the expertise of geodynamicists, geophysicists, Earth surface and atmospheric scientists.

Several geodynamic numerical models constrained by both geological and geophysical observations have been developed and some of these global models (Zhong et al., 2000; Moresi et al., 2003; Heister et al., 2017) have shown how mantle convection drives the motion of tectonic plates and dictates the long-term evolution of the Earth. Similarly progresses in the understanding of past, present, and future climates have been made by the development of mathematical models of the general circulation of a planetary atmosphere or ocean that simulate climate at an increasing level of detail (Dutkiewicz et al., 2016; Brown et al., 2018).

Yet, we are still missing a tool to evaluate global scale evolution of Earth surface and its interaction with the atmosphere, the hydrosphere, the tectonic and mantle dynamics. Such a tool will certainly provide new insights and help to better characterise many aspects of the Earth system ranging from the role of atmospheric circulation on physical denudation, from the influence of erosion and deposition of sediments on mantle convection, from the location and abundance of natural resources to the evolution of life.

The model presented in this paper is a first step toward the development of a parallel global scale landscape evolution model. It allows to couple the Earth's surface with global climatic perturbations and geodynamic forces acting within the Earth's interior. Landscapes and sedimentary basins evolution in eSCAPE are driven by a series of standard stream power incision and diffusion laws (Howard et al., 1994; Tucker and Slingerland, 1997; Chen et al., 2014) designed to address problems from regional to global scales and over geological time ($10^5$-$10^9$ years). Due to the inherent assumptions made in the set of equations used, eSCAPE is not intended to estimate the evolution of individual fluvial channels but to quantify large scale and long term evolution of Earth's surface regions (Salles et al., 2017; Armitage, 2019).

First, this paper presents the implicit, iterative approaches that are used to solve the multiple flow direction water routing and the erosion deposition processes (section 2). Then in section 3, I provide a list of all the parameters required to run the eSCAPE model and I discuss the input and output formats. In addition, three examples based on both generic and global scale experiments are described in detail and showcase the code main capabilities. Finally in section 4, I analyse the scalability of eSCAPE and discuss some of the limitations and future implementations that are necessary to improve the performance of the code on parallel architectures.

## 2 Modelled processes and algorithms

eSCAPE (Salles, 2018) is a parallel landscape evolution model, built to simulate landscapes and basins dynamic at various space and time scales over unstructured grids. The model accounts for river incision using stream power law, hillslope processes and sediment transport in land and marine environments. It can be forced with spatially and temporally varying tectonics (horizontal and vertical displacements) and climatic forces (temporal and spatial precipitation changes and sea-level fluctuations). eSCAPE is primarily written in Python with some functions in Fortran and takes advantage of PETSc solvers (Balay





et al., 2012) over parallel computing architectures using MPI. In this section, I describe the simulated physical processes along with the algorithms that are used.

## 2.1 Implicit parallel flow discharge implementation

Flow accumulation (FA) calculations are core component of landscape evolution models as they are often used as proxy to estimate flow discharge, sediment load, river width, bedrock erosion as well as sediment deposition. Until recently conventional FA algorithms were serial and limited to small spatial problems (O'Callaghan and Mark, 1984; Mark, 1988). With ever growing high resolution digital elevation dataset, new methods based on parallel approaches have been proposed over the last decade. Due to the recursive nature of FA computation, graph traversal techniques are common in determining the upstream-summation and most approaches (Wallis et al., 2009; Wallace et al., 2010; Tarboton, 2013; Bellugi et al., 2011; Braun and Willett, 2013) are based on an initial ordering process followed by efficient priority-queue implementations with some variants such as the sub-basin acyclic graph partitioning method in Salles and Hardiman (2016) or the breadth-first traversal approaches proposed by Barnes (2019). Except for the approach proposed by Barnes (2019), the previous methods scale well as long as the number of processors used is modest but quickly deteriorates as inter-processors communication cost increases.

In addition, when using the aforementioned implementation strategies, several problems might arise in (1) load balancing, when catchments size greatly changes in the simulated domain or (2) handling very high resolutions where multiple processes are needed for a single catchment. In addition, most of these methods assume a single flow direction (SFD - Fig. 1a). This assumption makes the emergent flow network highly sensitive to the underlying mesh geometry and most dendritic shape of obtained stream networks is often an artefact of the surface triangulation. To reduce this effect, authors have proposed to consider not only the steepest downhill direction but also to represent other directions appropriately weighted by slope (multiple flow direction - MFD). Using MFD algorithms prevent locking of erosion pathways along a single direction and help to route flow over flat regions into multiple branches (Tucker and Hancock, 2010). Yet, graph traversal approaches cannot be easily modified to incorporate MFD algorithms as catchments are no longer strictly isolated in low slope areas and flow pathways often diverge (Fig. 1b).

To overcome these limitations, Richardson et al. (2014) proposed to use linear solvers. The approach consists in writing the FA calculation as a sparse matrix system of linear equations (Eddins, 2007; Schwanghart and Kuhn, 2010). It can take full advantage of purpose-built, efficient linear algebra routines including those provided by parallel libraries such as PETSc (Balay et al., 2012). eSCAPE computes the flow discharge (m³/y) from FA and the net precipitation rate $P$ using the parallel implicit drainage area (IDA) method proposed by Richardson et al. (2014) but adapted to unstructured grids (Fig. 1).

The flow discharge at node $i$ ($q_i$) is determined as follows:

$$q_i = b_i + \sum_{d=1}^{N_d} q_d \tag{1}$$

where $b_i$ is the local volume of water $\Omega_i P_i$ where $\Omega_i$ is the voronoi area and $P_i$ the local precipitation value available for runoff during a given time step. $N_d$ is the number of donors with a donor defined as a node that drains into $i$ (as an example the donor of vertex 5 in the SFD sketch in Fig. 1a is 1). To find the donors of each node, the method consists in finding their





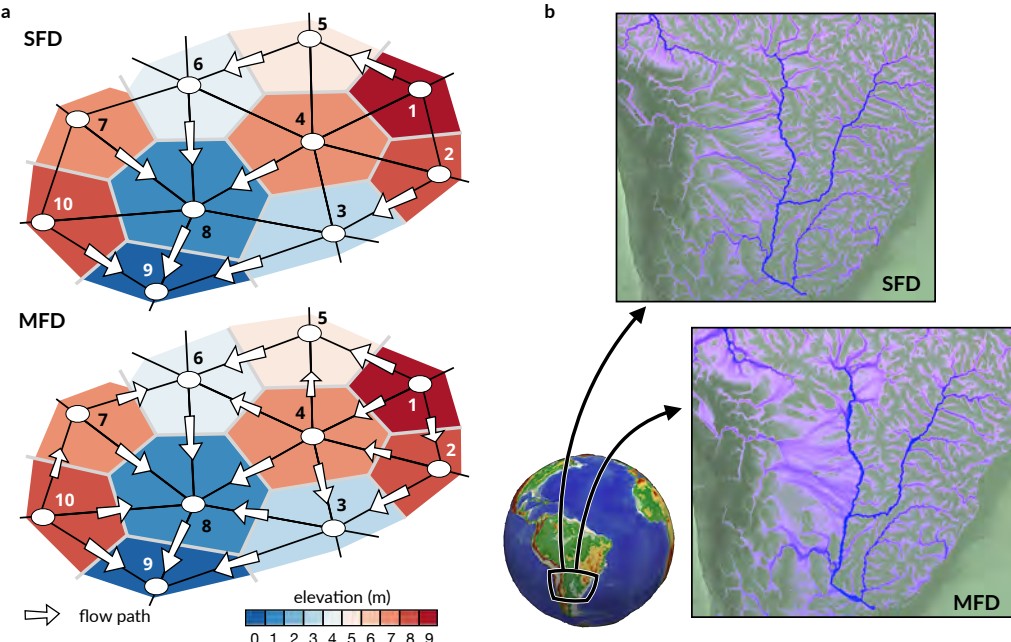

**Figure 1.** (a) Schematic diagram showing flow paths when considering a triangular irregular network composed of 10 vertices. Cells (i.e. voronoi area defining the region of influence of each vertex) are coloured by elevation. Two cases are presented considering single flow direction (top sketch – SFD) and multiple flow direction (bottom sketch – MFD/D$_\infty$). White arrows indicate flow direction and their sizes vary in proportion to slope (not at scale). Nodes numbers correspond to the subscripts in equations 2 and 4. (b) Differences in calculated drainage area for a portion of South America from eSCAPE using the two flow direction methods.

receivers first. Then, the receivers of each donor is saved into a receiver matrix, noting that the nodes, which are local minima, are their own receivers. Finally the transpose of the matrix is used to get the donor matrix. When Eq. 1 is applied to all nodes and considering the MFD case presented in Fig. 1a, the following relations are obtained:

$$
\begin{aligned}
q_i &= b_i \qquad i = 1,10 \\
q_2 &= b_2 + q_1 w_{1,2} \\
q_3 &= b_3 + q_2 w_{2,3} + q_4 w_{4,3} \\
q_4 &= b_4 + q_1 w_{1,4} + q_2 w_{2,4} \\
q_5 &= b_5 + q_1 w_{1,5} + q_4 w_{4,5} \\
q_6 &= b_6 + q_4 w_{4,6} + q_5 w_{5,6} + q_7 w_{7,6} \\
q_7 &= b_7 + q_{10} w_{10,7} \\
q_8 &= b_8 + q_3 w_{3,8} + q_4 w_{4,8} + q_6 w_{6,8} + q_7 w_{7,8} + q_{10} w_{10,8} \\
q_9 &= b_9 + q_3 w_{3,9} + q_8 w_{8,9} + q_{10} w_{10,9}
\end{aligned}
\tag{2}
$$





95  The choice of weights $w_{m,n}$ depends on the number of flow directions that is used. The weights range between zero and one and sum to one for each node:

$$\sum_n w_{m,n} = 1 \tag{3}$$

In eSCAPE, the number of flow direction paths is user-defined and can vary from 1 (i.e. SFD) up to 12 (i.e. MFD) depending of the grid neighbourhood complexity. The weights are calculated based on the number of downslope neighbours and are

100  proportional to the slope (Quinn et al., 1991; Tarboton, 1997; Richardson et al., 2014).

In matrix form the system defined in Eq. 2 is equivalent to **Wq=b** or:

$$
\begin{bmatrix}
1 & & & & & & & & & \\
-w_{1,2} & 1 & & & & & & & & \\
& & -w_{2,3} & 1 & -w_{4,3} & & & & & \\
-w_{1,4} & -w_{2,4} & & 1 & & & & & & \\
-w_{1,5} & & & -w_{4,5} & 1 & & & & & \\
& & & -w_{4,6} & -w_{5,6} & 1 & -w_{7,6} & & & \\
& & & & & 1 & & & -w_{10,7} \\
& & -w_{3,8} & -w_{4,8} & & -w_{6,8} & -w_{7,8} & 1 & & -w_{10,8} \\
& & -w_{3,9} & & & & & -w_{8,9} & 1 & -w_{10,9} \\
& & & & & & & & & 1
\end{bmatrix}
\begin{bmatrix}
q_1 \\ q_2 \\ q_3 \\ q_4 \\ q_5 \\ q_6 \\ q_7 \\ q_8 \\ q_9 \\ q_{10}
\end{bmatrix}
=
\begin{bmatrix}
b_1 \\ b_2 \\ b_3 \\ b_4 \\ b_5 \\ b_6 \\ b_7 \\ b_8 \\ b_9 \\ b_{10}
\end{bmatrix}
\tag{4}
$$

the vector **q** corresponds to the unknown flow discharge (volume of water flowing on a given node per year) and the elements of **W** left blank are zeros.

As explained in Richardson et al. (2014), the above system is implicit as the flow discharge for a given vertex depends on its neighbours unknown flow discharge. The matrix **W** is sparse and is composed of diagonal terms set to unity (identity matrix) and off-diagonal terms corresponding to at most the immediate neighbours of each vertex (typically lower than 6 in constrained Delaunay triangulation).

In eSCAPE, this matrix is built in parallel using compressed sparse row matrix functionality available from SciPy (Jones

et al., 2001). Once the matrix has been constructed, PETSc library is used to solve matrices and vectors across the decomposed domain (Balay et al., 2012). The performance of the IDA algorithm is strongly dependent on the choice of solver and preconditioner. In eSCAPE, the solution for **q** is obtained using the Richardson solver (Richardson, 1910) with block Jacobi preconditioning (*bjacobi*). This choice was made based on the convergence results from Richardson et al. (2014) but can be changed if better solver and preconditioner combinations are found. Iterative methods allow for an initial guess to be provided.

When this initial guess is close to the solution, the number of iterations required for convergence dramatically decreases. I take advantage of this option in eSCAPE by using the flow discharge solution from the previous time step as an initial guess. This allows to decrease the number of iterations of the IDA solver as discharge often exhibits small change between successive time intervals.





## 2.2 Erosion and sediment transport

River incision, associated sediment transport and subsequent deposition are critical elements of landscape evolution models. Commonly these are defined based on either a transport-limited (Willgoose et al., 1991) or a detachment-limited (Howard et al., 1994) approach. On one hand, the transport-limited hypothesis assumes that rivers may be able to transport sediment up to a concentration threshold (often referred to as the stream transport capacity) linked to discharge, slope, sediment size, and channel form (channel depth/width ratio) and that an infinite supply of sediment is available for transport. On the other

hand, the detachment-limited hypothesis supposes that erosion is not limited by a transport capacity but instead by the ability of rivers to remove material from the bed. Even though validations of each hypothesis have been conducted based on field studies calibration (Snyder et al., 2003; Tomkin et al., 2003; van der Beek and Bishop, 2003; Valla et al., 2010; Hobley et al., 2011) there are many evidences suggesting that both transport and detachment limited behaviours take place simultaneously in natural systems and models accounting for transition between the two have been proposed in the past (Beaumont et al., 1992;

Braun and Sambridge, 1997; Coulthard et al., 2002; Davy and Lague, 2009; Hodge and Hoey, 2012; Salles and Duclaux, 2015; Carretier et al., 2016; Turowski and Hodge, 2017; Lague, 2010; Shobe et al., 2017; Hobley et al., 2017; Salles et al., 2018). For simplicity, the approach proposed in this paper is similar to the initial version of eSCAPE (v1.0.0 - Salles (2018)) and is based on a standard form of the stream power law assuming detachment-limited only behaviour. In the future, a better representation of erosion and sediment transport could be added such as the SPACE approach proposed by Shobe et al. (2017).

As mentioned above and following Howard et al. (1994), I consider that sediment erosion rate is expressed using a stream power formulation function of river discharge and slope. The volumetric entrainment flux of sediment per unit bed area $E$ is of the following form:

$$E = KQ^m S^n \tag{5}$$

where $K$ is the sediment erodibility parameter, $Q$ is the water discharge, $S$ is the river slope. In eSCAPE, I incorporate

local precipitation-dependent effects on erodibility (Murphy et al., 2016) and use the flow discharge defined in previous section $Q = PA$ to represent rainfall gradients effect on discharge. $A$ is the flow accumulation and $P$ the upstream annual precipitation rate. $m$ and $n$ are scaling exponents. In our model, $K$ is user defined and the coefficients $m$ and $n$ are set to 0.5 and 1 respectively (Tucker and Hancock, 2010). $E$ is in $m/y$ and therefore the erodibility dimension is $(\text{m·y})^{-0.5}$.

The entrainment rate of sediment ($E$) is approached by an implicit time integration and consists in formulating the stream

power component in Eq. 5 in the following way:

$$\frac{\eta_i^{t+\Delta t} - \eta_i^t}{\Delta t} = -K\sqrt{Q_i} \frac{\eta_i^{t+\Delta t} - \eta_{rcv}^{t+\Delta t}}{\lambda_{i,rcv}}$$

where $\lambda_{i,rcv}$ is the length of the edges connecting the considered vertex to its receiver. Rearranging the above equation gives:

$$(1 + K_f)\eta_i^{t+\Delta t} - K_{f,i|rcv}\eta_{i,rcv}^{t+\Delta t} = \eta_i^t \tag{6}$$

with the coefficient $K_{f,i|rcv} = K\sqrt{Q_i}\Delta t/\lambda_{i,rcv}$. In matrix form the system defined in Eq. 6 is equivalent to: $\boldsymbol{\Gamma}\boldsymbol{\eta}^{t+\Delta t} = \boldsymbol{\eta}^t$.

Using the case presented in Fig. 1a, the matrix system based on the receivers distribution is defined as:




$$\begin{bmatrix} \gamma_{1,1} & -\gamma_{1,2} & & -\gamma_{1,4} & -\gamma_{1,5} & & & & & \\ & \gamma_{2,2} & -\gamma_{2,3} & -\gamma_{2,4} & & & & & & \\ & & \gamma_{3,3} & & & & & -\gamma_{3,8} & -\gamma_{3,9} & \\ & & -\gamma_{4,3} & \gamma_{4,4} & -\gamma_{4,5} & -\gamma_{4,6} & & -\gamma_{4,8} & & \\ & & & & \gamma_{5,5} & -\gamma_{5,6} & & & & \\ & & & & & \gamma_{6,6} & & -\gamma_{6,8} & & \\ & & & & & -\gamma_{7,6} & \gamma_{7,7} & -\gamma_{7,8} & & \\ & & & & & & & \gamma_{8,8} & -\gamma_{8,9} & \\ & & & & & & & & 1 & \\ & & & & & & -\gamma_{10,7} & -\gamma_{10,8} & -\gamma_{10,9} & \gamma_{10,10} \end{bmatrix} \begin{bmatrix} \eta_1^{t+\Delta t} \\ \eta_2^{t+\Delta t} \\ \eta_3^{t+\Delta t} \\ \eta_4^{t+\Delta t} \\ \eta_5^{t+\Delta t} \\ \eta_6^{t+\Delta t} \\ \eta_7^{t+\Delta t} \\ \eta_8^{t+\Delta t} \\ \eta_9^{t+\Delta t} \\ \eta_{10}^{t+\Delta t} \end{bmatrix} = \begin{bmatrix} \eta_1^{t} \\ \eta_2^{t} \\ \eta_3^{t} \\ \eta_4^{t} \\ \eta_5^{t} \\ \eta_6^{t} \\ \eta_7^{t} \\ \eta_8^{t} \\ \eta_9^{t} \\ \eta_{10}^{t} \end{bmatrix} \tag{7}$$

with

$$
\begin{aligned}
\gamma_{i,j} &= w_{i,j} K_{f,i|j} \qquad i \neq j \\
\gamma_{1,1} &= 1 + \sum_{j=2,4,5} w_{1,j} K_{f,1|j} \\
\gamma_{2,2} &= 1 + \sum_{j=3,4} w_{2,j} K_{f,2|j} \\
\gamma_{3,3} &= 1 + \sum_{j=8,9} w_{3,j} K_{f,3|j} \\
\gamma_{4,4} &= 1 + \sum_{j=3,5,6,8} w_{4,j} K_{f,4|j} \\
\gamma_{5,5} &= 1 + w_{5,6} K_{f,5|6} \\
\gamma_{6,6} &= 1 + w_{6,8} K_{f,6|8} \\
\gamma_{7,7} &= 1 + \sum_{j=6,8} w_{7,j} K_{f,7|j} \\
\gamma_{8,8} &= 1 + w_{8,9} K_{f,8|9} \\
\gamma_{10,10} &= 1 + \sum_{j=7,8,9} w_{10,j} K_{f,10|j}
\end{aligned}
\tag{8}
$$

This system is implicit and the matrix is sparse. The SciPy compressed sparse row matrix functionality (Jones et al., 2001) is used to build $\Gamma$ on local domains and the Eq. 7 is then solved using Richardson solver with block Jacobi preconditioning (*bjacobi*) using an initial guess for the solution set to vertices elevation.

Once the entrainment rates have been obtained, the sediment flux moving out at every node $Q_s^{out}$ equals the flux of sediment flowing in plus the local erosion rate. $Q_s^{out}$ takes the following form:

$$Q_s^{out} = Q_s^{in} + (1 - F_f) E \Omega$$

$\Omega$ is the voronoi area of the considered vertex and $F_f$ is the fraction of fine sediment that remains in suspension. The solution of the above equation requires the calculation of the incoming sediment volume from upstream nodes $Q_s^{in}$. At node $i$, Eq. 8 is





equivalent to:

$$q_{s,i} = e_i + \sum_{d=1}^{N_d} q_{s,d} \tag{9}$$

where $e_i = (1 - F_f)E_i\Omega_i$ and $N_d$ the number of donors. Assuming that river sediment concentration is distributed in a similar way as the water discharge we can write a similar set of equalities as the ones in Eq. 2. Then a matrix system as proposed for the FA (Eq. 4) can be obtained. The new system is then solved using the PETSc solver and preconditioner previously defined.

### 2.3 Priority-flood depression filling

In most landscape evolution models, internally-draining regions (e.g., depressions and pits) are usually filled before the calculation of flow discharge and erosion-deposition rates. This ensures that all flows conveniently reach the coast or the boundary of the simulated domain. In models intended to simulate purely erosional features, such depressions are usually treated as transient features and often ignored. However, eSCAPE is designed to not only address erosion problems but also to simulate source-to-sink transfer and sedimentary basins formation and evolution in potentially complex tectonic settings. In such cases, depressions may be formed at different periods during runtime and may be filled or remain internally drained (e.g., endorheic basins) depending on the volume of sediment transported by upstream catchments.

Depression filling approaches have received some attention in recent years with the development of new and more efficient algorithms (Wang and Liu, 2006; Barnes et al., 2014; Zhou et al., 2016, 2017; Wei et al., 2018). These methods based on priority-flood offer a time complexity of the order of $O(Nlog(N))$ compared to older approaches such as the Jenson and Domingue (1988) ($O(N^2)$) or Planchon and Darboux (2002) ($O(N^{1.2})$) algorithms.

Priority-flood algorithms consist in finding the minimum elevation a cell needs to be raised to (e.g., spill elevation of a cell) to prevent downstream ascending path to occur. They rely on priority queue data structure used to efficiently find the lowest spill elevation in a grid. Depending on the chosen method, priority queue implementation approaches affect the time complexity of the algorithm (Barnes et al., 2014). In eSCAPE, the priority-flood + $\epsilon$ variant of the algorithm proposed in Barnes et al. (2014) is implemented. It provides a solution to remove automatically flat surfaces and it produces surfaces for which each cell has a defined gradient from which flow directions can be determined.

In eSCAPE, this part of the algorithm is not parallelised and is performed on the master processor. It starts from the grid border vertices and processes vertices that are in their immediate neighbourhoods one by one in the ascending order of their spill elevations (Barnes et al., 2014). The initialisation step consists in pushing all the edges nodes onto a priority queue. The priority queue rearranges these nodes so that the ones with the lowest elevations in the queue are always processed first.

To track nodes that have already been processed by the algorithm a Boolean array is used in which edge nodes (that are by definition at the correct elevation) are marked as solved. The next step consists in removing (i.e. popping) from the priority queue the first element (i.e. the lowest node). This node $n$ is guarantee to have a non-ascending drainage path to the border of the domain. All non-processed neighbours (based on the Boolean array) from the popped node are then added to the priority queue. In the case where a neighbour $k$ is at a lower elevation than $n$ its elevation is raised to the elevation of $n$ plus $\epsilon$ before





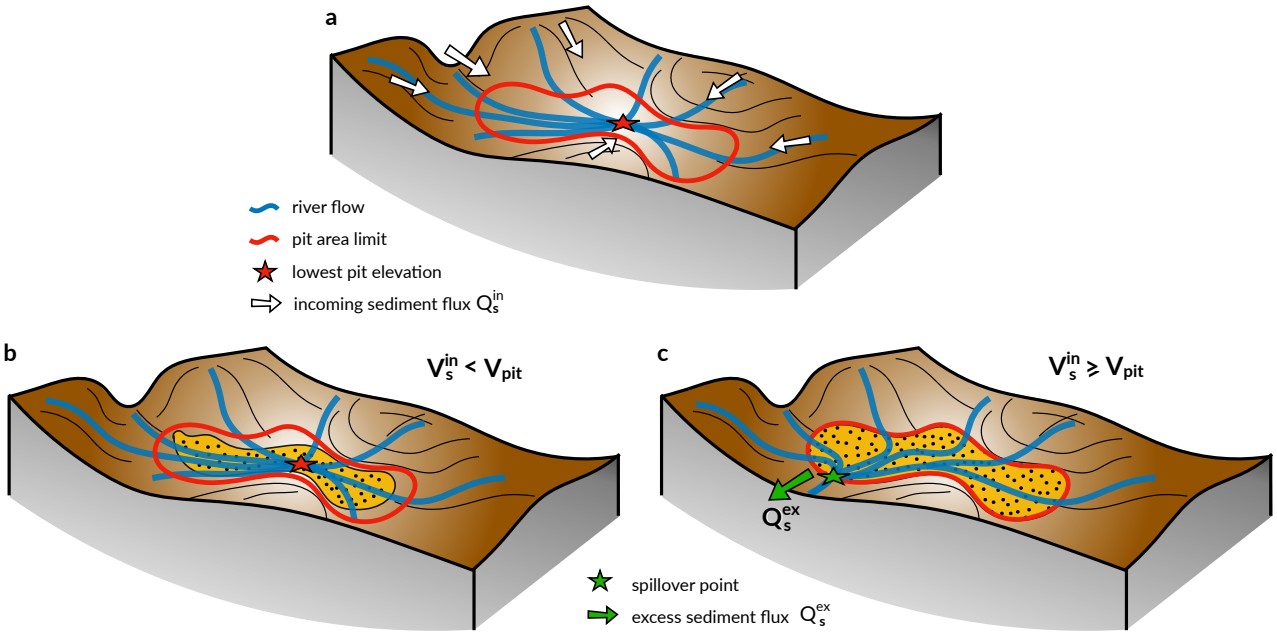

**Figure 2.** Illustration of the two cases that may arise depending on the volume of sediment entering an internally drained depression (panel a). The red line shows the limit of the depression at the minimal spillover elevation. b) The volume of sediment ($V_s^{in}$) is lower than the depression volume $V_{pit}$. In this case all sediments are deposited and no additional calculation is required. c) If $V_s^{in} \geq V_{pit}$, the depression is filled up to depression filling elevation (priority-flood + $\epsilon$), the flow calculation needs to be recalculated and the excess sediment flux ($Q_s^{ex}$) is transported to downstream nodes.

being pushed to the queue. Once $k$ has been added to the queue, it is marked as resolved in the Boolean array. In this basic
implementation of the priority-flood algorithm, the process continues until the priority queue is empty (Barnes et al., 2014).

### 2.4 Depression filling and marine sedimentation

The filling algorithm presented above is used to calculate the volume of each depression at any time step. Once the volumes of these depressions are obtained, their subsequent filling is dependent of the sediment fluxes calculation defined in section 2.2 (Fig. 2a). In cases where the incoming sediment volume is lower than the depression volume (Fig. 2b), all sediments are
deposited and the elevation at node $i$ in the depression is increased by a thickness $\delta_i$ such that:

$$\delta_i = \Upsilon(\eta_i^f - \eta_i) \tag{10}$$

where $\eta_i^f$ is the filling elevation of node $i$ obtained with the priority-flood + $\epsilon$ algorithm and the ratio $\Upsilon$ is set to $V_s^{in}/V_{pit}$.
If the cumulative sediment volume transported by the rivers draining in a specific depression is above the volume of the depression ($V_s^{in} \geq V_{pit}$ - Fig. 2c) the elevation of each node $i$ is increased to its filling elevation ($\eta_i^f$) and the excess sediment
volume is allocated to the spillover node (Fig. 2c). The updated elevation field is then used to compute the flow accumulation





following the approach presented in section 2.1. The sediment fluxes are initially set to zero except on the spillover nodes and using Eq. 9 the excess sediments are transported downstream until all sediments have been deposited in depressions, have entered the marine environment, or have moved out of the simulation domain.

In the marine realm, sedimentation computation follows a different approach to the one described above. First, the flow accu-
mulation is computed using the filled elevation in both the aerial and marine domains and a maximum volumetric deposition rate $\zeta_i$ is calculated based on the depth of each marine node:

$$\zeta_i = 0.9(\eta_{sl} - \eta_i)\Omega_i/\Delta t$$

with $\eta_{sl}$ the sea-level position. Using similar solver and preconditioner as the ones proposed for the flow discharge calculation, we solve implicitly a matrix system equivalent to the one in Eq. 4 with the same weight ($W$) and a vector $b$ equals to $q_{s,i} - \zeta_i$.
From the solution, only positive sedimentation rates are initially kept and the sedimentation thicknesses for these nodes are set to $\zeta\Delta t$. Then remaining sediment fluxes on adjacent vertices are found by computing the sum of $\zeta$ and obtained sedimentation rates and by considering again only positive values.

## 2.5 Hillslope processes and marine top sediment layer diffusion

Hillslope processes are known to strongly influence catchment morphology and drainage density and several formulations of
hillslope transport laws have been proposed (Culling, 1963; Tucker and Bras, 1998; Perron and Hamon, 2012; Howard et al., 1994; Fernandes and Dietrich, 1997; Roering et al., 1999, 2001). Most of these formulations are based on a mass conservation equation and with some exceptions such as CLICHE model (Bovy et al., 2016), these models assume that a layer of soil available for transport is always present (i.e. precluding case of bare exposed bedrock) and that dissolution and mass transport in solution can be neglected (Perron and Hamon, 2012).
Under such assumptions and via the Exner's law, the mass conservation equation widely applied in landscape modelling is of the form (Dietrich et al., 2003; Tucker and Hancock, 2010):

$$\frac{\partial \eta}{\partial t} = -\nabla \cdot q_{ds} \tag{11}$$

where $q_{ds}$ is the volumetric soil flux of transportable sediment per unit width of the land surface. In its simplest form, $q_{ds}$ obeys the Culling model (Culling, 1963) and hypothesises a proportional relationship to local hillslope gradient (i.e. $q_{ds} = -D\nabla\eta$,
also referred to as the creep diffusion equation):

$$\frac{\partial \eta}{\partial t} = D\Delta\eta \tag{12}$$

in which $D$ is the diffusion coefficient that encapsulates a variety of processes operating on the superficial soil layer. As an example, $D$ may vary as a function of substrate, lithology, soil depth, climate and biological activity (Tucker et al., 2001; Tucker and Hancock, 2010). The creep law is found in many models such as GOLEM (Tucker and Slingerland, 2017), CHILD (Tucker
and Slingerland, 1997), LANDLAB (Hobley et al., 2017) or Badlands (Salles and Hardiman, 2016; Salles et al., 2018), and in Willgoose et al. (1991), Fernandes and Dietrich (1997), Tucker and Slingerland (1997), Simpson and Schlunegger (2003).





In eSCAPE, hillslope processes rely on this approximation even though field evidences suggest that the creep approximation (Eq. 12) is only rarely appropriate (Roering et al., 1999; Tucker and Bradley, 2010; Foufoula-Georgiou et al., 2010; DiBiase et al., 2010; Larsen and Montgomery, 2012; Grieve et al., 2016a). In the future, a possible improvement could be based on the nonlinear hillslope transport equation incorporating a critical slope to model hillslope soil flux (Roering et al., 1999, 2001).

For a discrete element, considering a node $i$ the implicit finite volume representation of Eq. 12 is:

$$\frac{\partial \eta_i}{\partial t} = \frac{\eta_i^{t+\Delta t} - \eta_i^t}{\Delta t} = D \sum_{j=1}^{N} \frac{\chi_{i,j}(\eta_j^{t+\Delta t} - \eta_i^{t+\Delta t})}{\Omega_i \lambda_{i,j}} \tag{13}$$

$N$ is the number of neighbours surrounding node $i$, $\Omega_i$ is the voronoi area, $\lambda_{i,j}$ is the length of the edge connecting the considered nodes and $\chi_{i,j}$ is the length of voronoi face shared by nodes $i$ and $j$. Applied to the entire domain, the equation above can be rewritten as a matrix system $\boldsymbol{Q}\boldsymbol{\eta}^{t+\Delta t} = \boldsymbol{\eta}^t$ where $\boldsymbol{Q}$ is sparse. The matrix terms only depend on the diffusion coefficient $D$, the grid parameters and voronoi variables ($\chi_{i,j}$, $\lambda_{i,j}$, $\Omega_i$). In eSCAPE, these parameters remain fixed during a model run and therefore $\boldsymbol{Q}$ needs to be created once at initialisation. At each iteration, hillslope induced changes in elevation $\boldsymbol{\eta}$ are then obtained in a similar way as for the solution of the other systems using PETSc Richardson solver and block Jacobi preconditioning.

In addition to hillslope processes, a second type of diffusion is available in eSCAPE and consists in distributing freshly deposited marine sediments in deeper regions. This process is the only one treated explicitly and in this case the length of the diffusion time step $\Delta t_m$ must be less than a CFL factor to ensure numerical stability:

$$\Delta t_m < 0.1 \min_{i,j} \left( \lambda_{i,j}^2 / D_m \right) \tag{14}$$

where $D_m$ is the diffusion coefficient for the newly deposited marine sediments. Even with a reasonable small time step, the Eq. 14 can produce incorrect results. Following Bovy et al. (2016), the following set of inequalities are also added:

$$
\begin{aligned}
\Delta t \sum_j \chi_{i,j} q_{ms,ij}^{out} &\leq h_i \Omega_i \\
&\leq \alpha(\eta_i - \eta_m)\Omega_i
\end{aligned}
\tag{15}
$$

where $q_{ms,ij}^{out}$ is the flux of sediment from the marine top layer leaving node $i$ towards the downstream neighbours $j$, $h_i$ is the depth of the marine top layer, $\eta_m$ is the elevation associated to the highest downslope neighbour of $i$ and $\alpha$ is a factor lower than 1. These inequalities are always satisfied if positive outgoing fluxes are scaled by a factor $\beta$ given by:

$$\beta_i = \min\left( \frac{\Omega_i \min(h_i, \alpha(\eta_i - \eta_m))}{\Delta t \sum_j \chi_{i,j} q_{ms,ij}^{out}}, 1 \right) \tag{16}$$

In eSCAPE, marine diffusion of freshly deposited sediment is performed explicitly using the CFL condition described in Eq. 14 and the restriction proposed in Eq. 16.

## 3 Usability and applications

In this section, I present the main files used to run eSCAPE and to visualise the generated outputs. I then illustrate the capability of the code using a series of 3 examples presenting two generic models and one global scale experiment.





**Table 1.** Input parameters relative to intial surface, temporal extent and output.

| Parameters | Definition | Default values |
|---|---|---|
| **name** | Description of simulation - string | optional |
| **domain** | Definition of the simulated region | required |
| **filename** | TIN grid (vtk file) and elevation field - list | required |
| **flowdir** | Flow direction method integer between [1,12] | 1 |
| **bc** | Boundary conditions (choices: flat, fixed or slope) | slope |
| **sphere** | Set to 1 for spherical experiments | 0 |
| **time** | Simulation time definition all values are in years | required |
| **start** | Simulation start time | required |
| **end** | Simulation end time | required |
| **tout** | Simulation output interval time | required |
| **dt** | Simulation time step | required |
| **output** | Output folder | optional |
| **dir** | Directory name containing Hdf5, XMF and XDMF outputs | optional - Default name: output |
| **makedir** | Boolean is False: output folder with same name is deleted or | True |
| | Boolean is True: Keep previous folder adding a number | |

## 3.1 Input parameters and visualisation

eSCAPE uses YAML syntax for its input file. YAML structure is done through indentation (one or more spaces) and sequence items are denoted by a dash. When parsing the input file, the code is searching for some specific keywords defined in tables 1, 2 and 3. Some parameters are optional and only need to be set when specific forces (table 2) or physical processes (table 3) are
applied to a particular experiment.

All the input parameters that are defined in external files like the initial surface, different precipitation or displacement maps are read from VTK files. These input files are defined on an irregular triangular grid (TIN). Examples on how to produce these files are provided in the eSCAPE demo repository on Github and Docker. The only exception is the sea-level file which is a two-columns CSV file containing on the first column the time in years and ordered in ascending order and on the second one
the relative position of the sea-level in metres (**curve** in table 2).

The **domain** and **time** keywords (table 1) are required for any simulation. The flow direction method to be applied in a given simulation is specified with **flowdir** and takes an integer value ranging between 1 (for SFD) and 12 (for MFD/D$_\infty$). On the edges of the domain three types of boundary conditions (**bc**) are available and applied to all edges *flat*, *fixed* or *slope*. The *flat* option assumes that all edges elevations are set to the elevations of their closest non-edge vertices, the *fixed* option is used
when edges elevations need to remain at their initial positions during the model run and the *slope* option defines a slope based on the closest non-edge node average slope.





**Table 2.** Input parameters relative to forcing conditions.

| Parameters | Definition | Default values |
|---|---|---|
| **sea** | Sea-level forcing | optional |
| **position** | Relative sea-level position (m) | 0. |
| **curve** | File containing 2 columns (time and sea-level position) | optional |
| **climate** | Sequence of precipitation events in m/yr | optional |
| **start** | Starting time of a given event in year | required if module turned on |
| **uniform** | Either an uniform value or | 0. |
| **map** | A VTK map of spatial change in precipitation | |
| **tectonic** | Sequence of vertical tectonic events in m/yr | optional |
| **start** | Starting time of a given event in year | required if module turned on |
| **step** | Time step to apply tectonic time step in year | optional when **sphere**=1 |
| **end** | Ending time of a given event in year | optional when **sphere**=1 |
| **uniform** | Either an uniform value applied to all domain except edges or | 0. |
| **mapX** | Displacement VTK maps along esch axis defined either | |
| **mapY** | as rate in m/yr if the **sphere** parameter is set to 0 or | |
| **mapZ** | as a distance in m if **sphere**=1 | |

**Table 3.** Input parameters relative to physical processes.

| Parameters | Definition | Default values |
|---|---|---|
| **sp_br** | Stream power parameters for bedrock | required |
| **Kbr** | Bedrock erodibility ($m^{-0.5}yr^{-0.5}$) | $1.e^{-12}$ |
| **sp_dep** | Deposition parameter definition | optional |
| **Ff** | Fraction of sediment in suspension [0.,1.] | 0. |
| **diffusion** | Diffusion parameters declaration | optional |
| **hillslopeK** | Hillslope diffusion coefficient ($m^2$/yr) | required |
| **sedimentK** | Marine fresh sediment coefficient ($m^2$/yr) | 10. |

The **climate** and **tectonic** keywords (table 2) may be defined as a sequence of multiple forcing conditions each requiring a starting time (**start** in years) and either a constant value applied to the entire grid (**uniform**) or spatially varying values specified in a file (**map**).

Surface processes parameters (table 3) define the coefficients for the stream power law (**Kbr** is $K$ in Eq. 5). It is worth noting that the coefficient $m$ and $n$ are fixed in this version of eSCAPE and take the value 0.5 and 1 respectively. The **diffusion** keyword defines both hillslope (creep law – Eq. 12 and **hillslopeK** is $D$ in $q_{ds} = -D\nabla\eta$) and marine diffusion coefficients. The freshly deposited marine sediments are transported based on a diffusion coefficient **sedimentK** equivalent to $D_m$ in $q_{ms} = -D_m\nabla\eta$ and used in Eq. 13 with the restriction proposed in Eq. 16.





The model outputs are located in the output folder (**dir** keyword – table 1) and consist of a time series file named eSCAPE.xdmf
and two additional folders (h5 and xmf). The HDF5 files are wrote individually for each processors and the XMF files combine
them together to show the global outputs. The XDMF file is the main entry point for visualising the outputs and should be
sufficient for most users. The file can be opened with the Paraview software (Ahrens et al., 2014).

### 3.2   Examples

#### 3.2.1   Analysing the influence of time step on eSCAPE runs

The first example illustrates the effect of increasing time step length on the resulting landscape evolution. The initial surface
consists in a flat triangulated squared grid of 100 km side and approximately 100 m resolution containing $\simeq$1.3 million points.
This surface is exposed to an uniform precipitation regime of 1 m/y and is uplifted linearly from its fixed western side to the
eastern one that experiences an uplift of 5 mm/y (Fig. 3a). The proposed setting is similar to the one in Braun and Willett (2013)
and the value of the bedrock erodibility parameter $K$ is set to $2 \times 10^{-4}$ in order to reach steady-state during the simulated $10^5$
years. Under such conditions, the model is purely erosional and therefore neither the aerial and marine sedimentation nor the
depression filling algorithm are considered. In addition hillslope processes are also turned off, meaning that this example only
relies on the implicit parallel flow discharge and erosion equations defined in sections 2.1 and 2.2.

Three cases are presented after $10^5$ years for different time steps $\Delta t$ varying from $10^4$ to $10^3$ and $10^2$ years in Figure 3a,
3b and 3c respectively, implying that the number of steps is 10, 100 and 1000. In both cases the implicit schemas converge
for the chosen solver and preconditioner (*i.e.* Richardson with block Jacobi). The solutions for the mean landscape elevation
(Fig. 3d) show that the landscape reaches steady state in all cases and overall the final elevations are in good agreement with
a maximum elevation of $482 \pm 3$ m and a number of catchments $n_c$ almost identical between models ($87 \leq n_c \leq 94$). Yet as
the time step increases the differences between models increase over time. By the end of the simulation, the mean elevation
difference between the case with $\Delta t$ equals to $10^2$ y and the one at $10^3$ y is around $2.5\%$ whereas the difference with a $\Delta t$
of $10^4$ y is above $30\%$ (Fig. 3d). It illustrates the transient nature of the landscape and its strong dependence to antecedent
morphologies. Even small changes on elevation could potentially trigger completely different landscape features. Compared
to the explicit algorithm proposed for the drainage area computation in Braun and Willett (2013), the approach here relies on
an implicit schema and produces a more stable solution for longer time scale. Yet time step limitations are still required to
ensure a good representation of landscape features (*e.g.* knickpoint propagation) and care should be taken when choosing a
given simulation time step.

As mentioned in section 2.1, the iterative linear solvers of the implicit methods for both flow accumulation and erosion use
previous time step solution as an initial guess. In cases where the landscape does not change significantly between consecutive
time steps, both the flow accumulation and erosion rates are likely to remain almost unchanged and the number of iterations
required by the solver to reach convergence will be small. As an example if the drainage network remains the same between
two iterations, the flow accumulation solver solution will be obtained immediately and the results given directly. It highlights
a second implication of the choice of time step. Not only does the time step influences the final landscape morphology, it



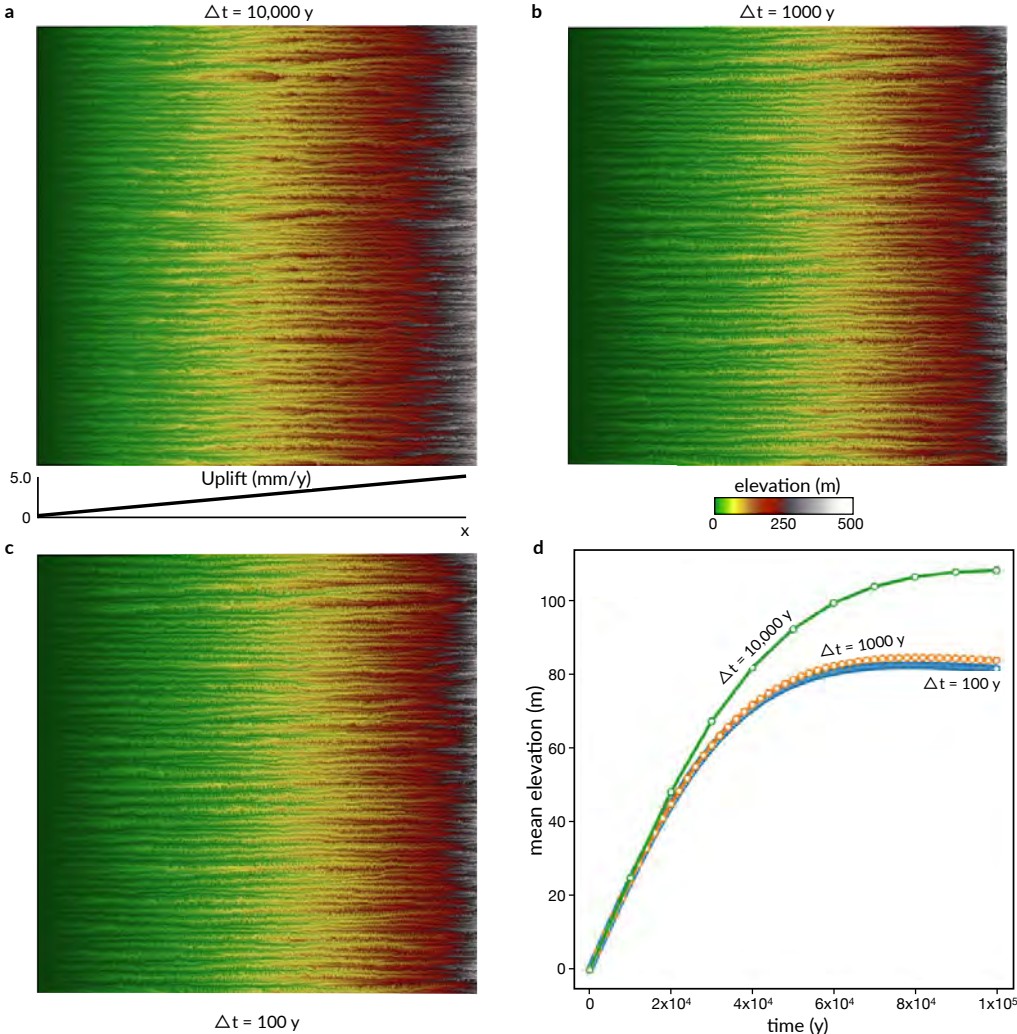

**Figure 3.** Resulting topographies of an initial flat squared surface (100 km side) after 100,000 years of uniform precipitation and linear uplift from west to east. Three simulations are performed in which the time step $\Delta t$ is set to a) $10^4$, b) $10^3$ and c) $10^2$ years. Panel d) presents the temporal change in mean elevation for the three cases. Differences between the runs are related to the transient nature of landscape evolution.

also controls the model running time. In some cases, similar running times will be achieved with smaller time steps if solvers solutions are obtained in a reduced number of iterations.

### 3.2.2 Comparison of single and multiple flow direction algorithms

In this second example, I present a series of three experiments in which the flow routing calculations are based on one (SFD), two and multiple (MFD) flow direction approaches (Fig. 4). In eSCAPE, it is possible to use different flow-routing algorithms

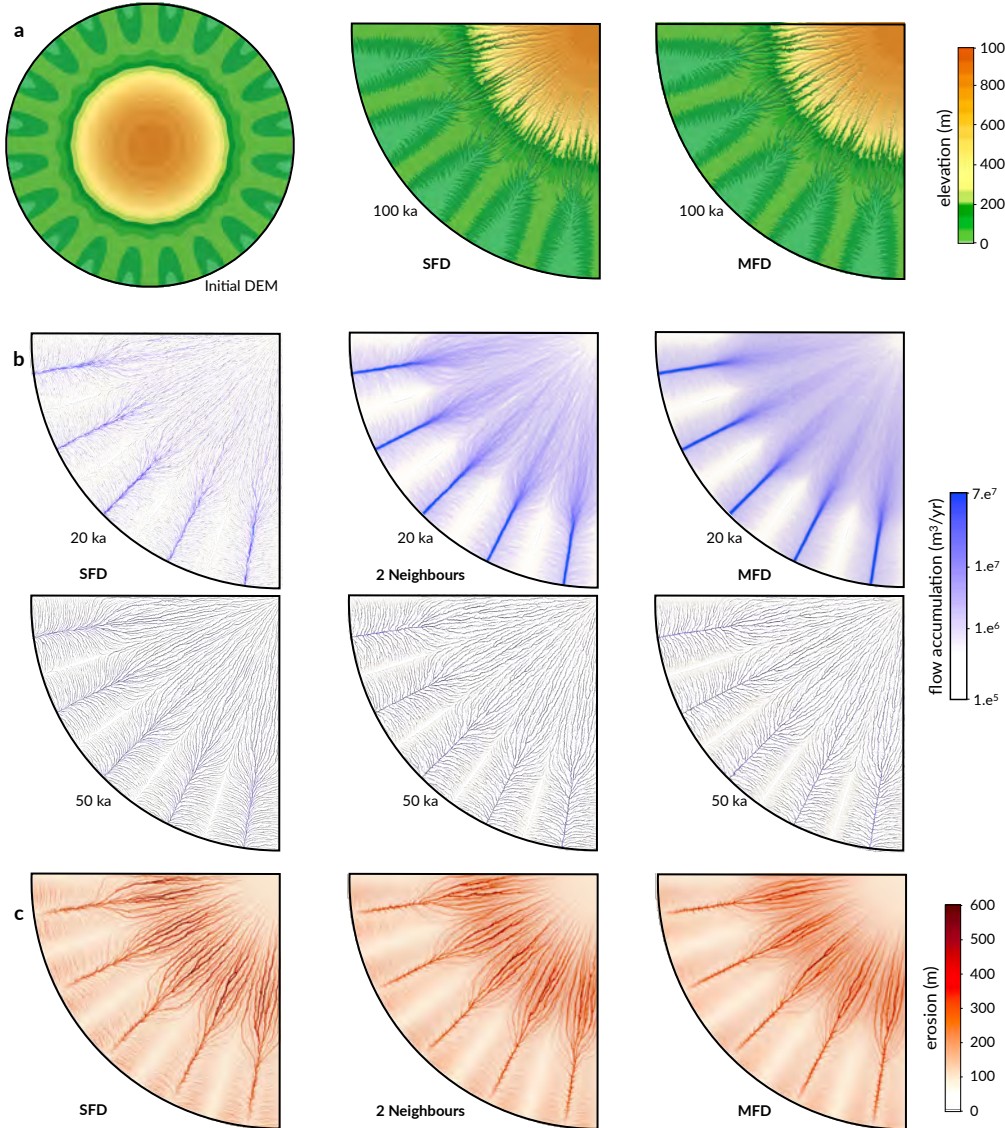

**Figure 4.** Effect of flow routing algorithms on flow accumulation patterns and associated erosion. Panel a) presents the initial radially symmetric surface defined with a central, high region and a series of distal low-lying valleys. Resulting topographies of the south-west area after 100,000 years of evolution under uniform precipitation for the SFD and MFD algorithms are shown on the right hand-side. Patterns of flow accumulation after 20,000 and 50,000 years for the SFD, two neighbours and MFD approaches are presented in panel b) as well as estimated landscape erosion at the end of the simulation time interval are given in panel c).

by specifying the number of directions (Fig. 1a and **flowdir** parameter in table 1) appropriately weighted by slope that rivers could potentially take when moving downhill.





For this example, the initial surface consists in a rotationally symmetric surface (Fig 4a) composed of valleys and ridges with lowest regions (at 0 m elevation) located on the edges of the domain and increasing to 1000 metres towards the center. The triangulated circular grid of 50 km radius is built with a resolution of approximately 200 m. The three experiments with varying water routing directions are ran for 100,000 years with a $\Delta t$ of 1000 years under a 1 m/y uniform precipitation. In addition to stream incision (bedrock erodibility $K$ set to $2 \times 10^{-5}$), hillslope processes are also accounted for using a diffusion coefficient

$D$ of $10^{-2}$ m$^2$/y.

After 20,000 years, the dendritic flow accumulation pattern observed on the surface for the SFD case (Fig. 4b) is analogue to many natural forms of drainage systems but is actually a numerical artefact and depends on the random locations of the nodes in the surface triangulation. By increasing the number of possible downstream directions, this sensitivity to the mesh discretisation is significantly reduced (as illustrated in Fig. 4b where a second direction is added). In addition, routing flow to

more than one destination node allows a better representation of channels pathways divergence into multiple branches over flat regions (Tucker and Hancock, 2010).

Landscape evolution models tend to be highly dependent on grid resolution and this dependency is mostly related to the approach used to route water down the surface (Schoorl et al., 2000; Pelletier, 2004; Armitage, 2019). As discussed by Armitage (2019), enabling *node-to-node* MFD algorithm decreases the dependence of landscape features (*e.g.* valley spacing, branching

of stream network, sediment flux etc) to grid resolution. As shown in Figure 4b and c, SFD algorithm leads to increase branching of valleys whereas the MFD approach, by promoting wider flow distribution, produces smoother topography where local carving of the landscape is reduced. Armitage (2019) also showed that when using models that operate at scale larger than river width resolution, *node-to-node* MFD algorithm creates landscape features that are not resolution dependent and that evolve closer to the ones observed in nature. Therefore it is recommended to use more than one downhill direction (**flowdir**)

in eSCAPE when looking at global and continental scale landscape evolution or for cases where multiple resolutions are considered within a given mesh.

### 3.2.3  Global scale simulation

The last example showcases a global scale experiment with eSCAPE. The simulation looks at the evolution of the Earth 200,000 years into the future starting with present day elevation and precipitation maps. The model is ran forward in time

without changing the initial forcing conditions and is primarily used to highlight the capabilities of eSCAPE and does not represent any particular geological situation.

The elevation is obtained from the ETOPO1 1 arc-minute global relief model of Earth's surface that integrates land topography and ocean bathymetry (Amante and Eakins, 2009). For the rainfall, I summed all the WorldClim gridded climate monthly dataset to obtained a global yearly rainfall map with a spatial resolution of about 1 km$^2$ (Fick and Hijmans, 2017). From

these dataset, I then built the initial surface and climate meshes at 16 km resolution consisting in approximately 3 millions points (Fig. 5a and 6a). The model inputs are temporally uniform, but any other climatic scenarios could have been chosen for illustration as well as tectonic conditions (both vertical: uplift and subsidence and horizontal: advection displacements).



**Figure 5.** eSCAPE global scale experiment of Earth morphological evolution over 200,000 years. Panel a) presents the initial elevation based on ETOPO1 dataset and forcing precipitation obtained from the WorldClim dataset. Panels b) and c) show the elevation and cumulative erosion-deposition resulting from the action of rivers and hillslope processes at different time intervals.





**Figure 6.** Similar to figure 5 from a different perspective.

In addition to these grids, the following parameters are chosen: **flowdir** is set to 5 (similar to the MFD flow routing approach), the time step $\Delta t$ equals 500 years, bedrock erodibility $K$ is $5 \times 10^{-5}$, the diffusion coefficient $D$ equals $10^{-1}$, the fraction of
sediment in suspension **Ff** is 0.3 ($F_f$ parameter in section 2.2 and defined in table 3) and the marine fresh sediment diffusion





$D_m$ is set to $5 \times 10^5$ (see section 2.5 and **sedimentK** parameter in table 3). The simulation took 2 hours to run on a cluster using 32 processors.

From these set of input parameters, eSCAPE can predict the global evolution of topography (Figures 5 and 6) and quantifies the associated volume and spatial distribution of sediments trapped in continental plains or transported into the marine realm.

By recording eroded sediment transport over each drainage basin, eSCAPE provides estimation of sedimentary mass fluxes carried by major rivers into the ocean (section 2.2). As shown in Figures 5 and 6, the predicted locations of largest basin outlets match quite well with observations and many of the biggest simulated deltaic systems are related to sediment transported by some of the world's largest rivers (Milliman and Syvitski, 1992; Syvitski et al., 2003). The model can also be used to evaluate the evolution of drainage systems, the stability of continental flow directions, the exhumation history of major mountain

ranges, the timing and geometry of sedimentary bodies formation (*e.g.* deltas or intra-continental deposits) as well as basins stratigraphy. All these predictions can be directly compared to sedimentary (sediment budgets, paleogeographic maps, etc) or thermochronology data.

This simulation illustrates a global scale model of Earth's surface evolution. In case where paleo-climatic conditions are known then eSCAPE can in principle be used to perform quantitative analysis of different tectonic forcings with complex spatial and

temporal variations. The results of these tests can then be compare with available geological records (such as denudation rates, paleotopographies, basins sedimentary thicknesses/volumes). In addition to climate and tectonic conditions, it is also possible to impose varying sea level fluctuations over geological times.

As such and even with the limited number of simulated processes, eSCAPE can be used to retrieved global sedimentary basins formation and their evolution based on temporal and spatial responses of both landscape and sediment fluxes to different sea

level conditions and tectonic and precipitation regimes.

## 4   Performance analysis

The performance of the implicit flow accumulation, erosion and sediment transport algorithms is strongly dependent on the choice of solver and preconditioner. As shown in sections 2.1 and 2.2, the forms of the matrices are not symmetric or positive definite and in this case only a limited number of iterative solvers and preconditioners are suitable. From the extensive

analysis provided in Richardson et al. (2014), the non-Krylov solver based on the Richardson method (Richardson, 1910) has been chosen in eSCAPE as it converged with the greatest number of preconditioners and exhibited superior scaling. For the preconditioner, several candidates (SOR, ILU, ASM...) are available (Saad, 2003) and I decided to use the block Jacobi preconditioner as it is one of the simplest methods and produces in combination with the Richardson solver good scalability (Richardson et al., 2014). Yet other combinations such as the Richardson solver with the Euclid preconditioner (*i.e.* HYPRE

package, Falgout et al. (2012)) might exhibit better scalability in some cases.

The analysis of the profiling work realised in Fig. 7a suggests that for purely erosive models (similar to the ones presented in section 3.2.1) most of the computational time is spent solving the Richardson iterative method (*PETSc solver KSP*). From the graph on the right hand side of the panel (Fig. 7a), one can deduce that performance improvements are obtained when the





problem size increases. However the scaling performance decreases when reaching 32/64 processors depending on the problem
size. This does not agree with some of the conclusions from Richardson et al. (2014), where the scaling of the implicit drainage
accumulation algorithm continues even for large numbers of processors (>192). To improve performance, I will be exploring
two directions. First the problem might be related to the chosen solver and preconditioner combination and I will run new tests
using the HYPRE package as discussed above. Secondly, the poor performance for larger processors might also be linked to
issues related to either Python PETSc wrapper or installation problems and incompatibilities between some of the compilers
and packages that I used. In the future, different software libraries and compilers versions from GNU and Intel will be tested
and might help to improve the performance for increasing number of processors.

For experiments accounting for marine deposition and pit filling, a similar trend is found when comparing performance against
processors number (right hand side in Fig. 7b). However the results from the profiling (left hand side in Fig. 7b) suggest
that more than half of the computation time is now spent on non-PETSc work with the biggest proportion related to the pit
filling function. In eSCAPE, the priority-flood algorithm is performed in serial (see section 2.3). This is the major limitation
of the code as shown by the time spent in broadcasting the information from the master node to the other processors (MPI
Bcast in Fig. 7b). To take advantage of parallel architectures, several authors (Wallis et al., 2009; Tesfa et al., 2011; Yıldırım
et al., 2015; Zhou et al., 2017) have proposed partitioning implementation of depression filling algorithms. However, most
of these methods require frequent interprocess communication and synchronisation. This becomes even more problematic in
the case of eSCAPE where the depression filling algorithm needs to be performed at every time step (Barnes, 2019). Barnes
(2017) presented an alternative to the aforementioned parallelisation methods that limits the number of communications. Yet
this approach is not fully satisfactory as it only fills the depressions up to the spilling elevation but does not provide a way
of implementing efficiently the $\epsilon$ variant of the algorithm proposed in (Barnes et al., 2014). Finding a strategy to perform a
parallel version of the $\epsilon$ variant of the priority-flood algorithm that could provide flow directions on flat surface will likely
improve greatly the performance of eSCAPE.

## 5 Conclusions

In this paper, I describe eSCAPE, an open-source, Python-based software designed to simulate sediment transport, landscape
dynamics and sedimentary basins evolution under the influence of climate, sea level and tectonics. In its current form, eSCAPE
relies on the stream power and creep laws to simulate the physical processes acting on the Earth's surface. The main differ-
ence with other landscape evolution models relies on the formulation used to solve the system of equations. The approach
builds upon the Implicit Drainage Area calculation from Richardson et al. (2014) and consists in a series of implicit iterative
algorithms for calculating multiple flow direction and erosion deposition that written in matrix form. As a result, the obtained
systems can be solved with widely available parallel linear solver packages such as PETSc.

Performance analysis shows good parallel scaling for small number of processors (under 64 processors as shown in section 4)
but some work is required for larger numbers. The code profiling suggests that the main issue is in the inter-processes commu-
nications happening when broadcasting the pit filling information computed in serial by the master to the other processors. In



**Figure 7.** Sunburst visualisation obtained from SnakeViz package showing the profiling results of multiple eSCAPE experiments. The analysis is performed for different numbers of processors (up to 256). On the left hand side, the fraction of time spent in each function is represented by the angular extent of the different arcs. On the right hand-side results of the computational runtime versus processors number over a series of time steps is given for experiments of different size. Panel a) presents the results for purely erosional simulations such as the ones presented in the first example (section 3.2.1). For panel b) eSCAPE is ran with all the processes turned on and uses a global scale experiments similar to the last example (section 3.2.3).



the future, a parallel approach allowing depression filling and flow direction computation over flat regions will be critical to improve the overall performance of the code.

Examples are provided in the paper and available through the Docker container. They illustrate the extent of temporal and
spatial scales that can be addressed using eSCAPE. As such this code is highly versatile and useful for geological applications related to source to sink problems at regional, continental and for the first time global scale. It is already possible to use eSCAPE to simulate global geological evolution of the Earth's landscape at about 1 km resolution providing accurate estimates of quantities such as large-scale erosion rates, sediment yields and sedimentary basins formation. In the future, the code will be coupled with atmospheric and geodynamic models to bridge the gap between local and global scales predictions of Earth
past and future evolutions.

*Code and data availability.* The source code with examples (Jupyter Notebooks) is archived as a repository on Github as the release version v2.0 from Zenodo (doi:10.5281/zenodo.3239569). The code is licensed under the GNU General Public License v3.0. The easiest way to use eSCAPE is via our Docker container (searching for **Geodels escape-docker** on Kitematic) which is shipped with the complete list of dependencies and the case studies presented in this paper. Our wiki page provides useful documentation regarding installation and code
usage. API documentation is available from the eSCAPE-API website.

*Author contributions.* T.S., development of the code, design of the experiment, output analysis, and manuscript writing.

*Competing interests.* The author declares that he has no conflict of interest.

*Acknowledgements.* The author acknowledge the Sydney Informatics Hub and the University of Sydney's high performance computing cluster Artemis for providing the high performance computing resources that have contributed to the research results reported within this
paper. The author also acknowledge the technical assistance of David Kohn from the Sydney Informatics Hub which was supported by Project #3789 and from Artemis HPC Grand Challenge.



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
