# Peer review of "eSCAPE: Regional to Global Scale Landscape Evolution Model v2.0"

_Geoscientific Model Development, 2019_

## Referee Comment (RC1) · John Armitage (Referee) · 1 Jul 2019

This manuscript presents "eSCAPE", which is a model developed to explore landscape erosion and deposition both above and below sea-level. The model is relatively well described and uses a nice range of techniques to model many of the sub-processes involved. I would like to hear how the author defends criticism that this model is not novel. eSCAPE v1 was published in the Journal of Open Source Software, how does v2 differ? What makes it require a whole new publication? Furthermore, and I ask this out of naivety, how does eSCAPE differ from Badlands? Is the difference significant? Overall this is my only major concern, and it is one that is potentially wrong. Otherwise in general the manuscript is well written (except for a few grammatical errors with verb conjugation).

[Figure]

Minor comments in the order in which the come in the text:

The introduction way oversells the model. Yes, it can model global erosion and deposition using a set of rules, however the model cannot capture lateral movement of the surface due to faulting. In fact there is no faulting, which is arguably the major process that connects mantle convection to surface processes. This is a very challenging problem, and not one the author seeks to solve. However, much text is wasted on describing a vision of a global coupled model. This should be saved for a research proposal and not used here. Explain what the advance is in this model, how it advances on v1, and Badlands. What is eSCAPE v2 for?

Line 15: What was the reason for cherry picking these citations, none of which date from the 80's?

Line 28: What is the purpose of this paragraph? As it is, it is far too short to encompass how global mantle flow is expressed at the earth's surface.

Line 70: I thought the approach of Jean Braun was O(N) efficient, always? Is the author saying otherwise?

Equation 2: The first line does not make sense. $q_1 = b_1$, not $q_i = b_i$

Line 127: "calibration" is out of place here.

Line 128: "evidence" should not get an "s", likewise "behaviour". There are other minor grammatical errors which I am sure will be corrected when copy edited.

Equations 7 and 8: Here it is hard coded that n=1 and m=0.5. This is stated later in the manuscript, but this is potentially a major limitation of the model, as the recent study by Kwang & Parker (2017) suggests that "the choice m∕n =0.5 yields a curiously unrealistic result: the predicted landscape is invariant to horizontal stretching".

[ Kwang, J. S. and Parker, G.: Landscape evolution models using the stream power incision model show unrealistic behavior when m ∕ n equals 0.5, Earth Surf.

[Figure]

Dynam., 5, 807-820, https://doi.org/10.5194/esurf-5-807-2017, 2017. ]

Line 159: In this equation the non-suspended sediment gets left behind, right? But the stream power law assumes instantaneous sediment transport. Therefore the two are incompatible? I am missing something here. Perhaps some additional explanation of how the model goes from erosion to deposition would help.

Section 2.3: The "priority-flood" algorithm is non-physical, right? I wonder if it should not be done after the hillslope processes (diffusion), as this would smooth depressions and potentially fill them. Then the subsequent filling by fluvial deposition should occur?

Section 2.5: Does marine deposition use a constant diffusion coefficient? Some marine deposition models vary this diffusion coefficient with water depth, to simulate wave and tide effects. I assume that this is not the case within eSCAPE?

Table 3: I think the marine parameters are missing from the table.

Summary:

I think eSCAPE is a very nice model but I think this manuscript could explain the model a little bit better. My only major criticism is the lack of clarity on how v2 of this model differs from v1, and why that difference warrants another publication.

Reproducibility:

The code is available and I have successfully installed it. I have come across minor issues in running the code, due to my install of python and petsc, but this will be fixed before publication I am sure.

I hope these comments help

John Armitage Institute de Physique du Globde de Paris

---

## Referee Comment (RC2) · Benoit Bovy (Referee) · 14 Aug 2019

General comments:

This paper presents the eSCAPE landscape evolution model. It is overall well written and nicely illustrated. eSCAPE features are explained in a consistent order and each with a sufficient level of detail. The manuscript contains references to comprehensive material available online that will help users installing, running eSCAPE and reproduce the experiments presented in this paper.

The author clearly highlights the various benefits of using a matrix-based approach for computing flow accumulation and sediment transport in both continental and marine environments. The most outstanding benefits are, in my opinion, the implicit scheme

used for computing drainage area, the horizontal scalability (distributed computing) and the ability to re-use powerful libraries like PETSc. This approach is not new, but eSCAPE represents to my knowledge the first effort towards integrating it into a user-friendly landscape evolution model that has the minimal set of features required for applications at global scale.

In the examples section, the author carefully shows the implications of the choice of the time step, even in the context of implicit numerical schemes used for most of the model components, which is very much appreciated.

eSCAPE is designed to handle very large grids and aims at simulating landscape evolution at regional to global scales. The author states that this is the first landscape evolution model able to simulate processes at global scale.

Although simulations at this scale are presented in the manuscript, the resolution (16 km) and the total number of mesh points (3 millions) used for these simulations remain somewhat coarse and small, respectively. The mesh size is not much greater than what could be processed nowadays at tractable computational costs using sequential model implementations. In this regard, it would be interesting (1) to see how eSCAPE performance does roughly compare with efficient, sequential implementation(s) of the same processes run on grids or meshes of a similar size (at least for the SFD / purely erosive case), and (2) to see how eSCAPE scales at much greater mesh sizes of, e.g., 10-100 millions of nodes (at least for the purely erosive case since applying pit resolving in serial might become computationally intractable at that scale).

In my opinion, and this is my main concern, the manuscript could do a better job at showing when eSCAPE would become a compelling alternative to landscape evolution models based on sequential algorithms (graph traversal or other). The two suggestions above might help improving that.

Besides that, I have a minor comment about the method chosen in eSCAPE to control the convergence vs. divergence of flow paths (SFD vs. MFD). Is there any specific

reason why manually setting the number of flow receivers is preferred over unconditionally partitioning the outgoing flow between all downslope neighbors and let the user control the SFD vs MDF behavior, e.g., by tuning the parameter(s) of the weight vs. slope relationship? While both approaches to this problem are arbitrary, the second one would allow finer control and has been studied more in depth (see, e.g., Qin et al. 2007, https://doi.org/10.1080/13658810601073240). Or perhaps a combination of the two approaches would allow even greater flexibility.

Specific comments:

Line 69: Due to the issue (rightly pointed by the author in the following paragraph) of load balancing vs. the relative sizes of the catchments in the simulated domain, methods based on depth-first graph traversal may not scale at all in the worst case scenario of one single simulated catchment.

Line 78: Note that approaches like the one described by Braun and Willet's (2013) can actually be easily modified to incorporate MFD algorithms. Unfortunately, no paper has been published yet on this.

Fig 1a: I guess that the color map used for cell elevation values has been chosen so that it emphasizes dry (high) vs. wet (low) cells. Still, I doubt that the value of 4 meters has a special meaning, and a non-diverging color map would be more appropriate.

Line 143: "m/y" units badly formatted.

Line 154: I might be missing something in the source code, but "scipy.sparse" is imported only in "surfprocplex.py" and it is not used further in that module. Maybe there is an inconsistency between the source code and the manuscript about how SciPy is used here?

Line 178: It might be worth also mentioning the O(N) depression resolving algorithm (not part of the "priority-flood" family of algorithms) that has been published very recently by Cordonnier et al. (2019, https://doi.org/10.5194/esurf-7-549-2019). Dis-

claimer: I'm co-author of this paper.

Section 2.4: It might be worth adding a few words on how the depression areas are delineated and how the spillover nodes are retrieved using the priority-flood + epsilon filling algorithm. This is not obvious, at least to me and potentially to other readers as well.

Line 380: Typo "be compare" -> "be compared".

Line 415: The Cordonnier et al.'s depression resolving algorithm cited here above is optimized specifically for use in landscape evolution models. Compared to the priority flood + epsilon variant of Barnes et al. (2014), it may drastically improve the overall performance when it is executed at every time step. That said, there is no parallel version yet and it works best when coupled with graph traversal algorithms.

Figure 7: Very nice figure that captures detailed profiling results at a glance!

---

## Author Comment (AC2) · 17 Aug 2019

**REVIEWER 1**

First, I would like to thank the reviewer for his useful comments. Below is my response to these comments.

**General comments:**

**GC1:** I would like to hear how the author defends criticism that this model is not novel. eSCAPE v1 was published in the Journal of Open Source Software, how does v2 differ? What makes it require a whole new publication?

**Response:** The main differences between v1 and v2 of eSCAPE are in the way v2 handles the marine deposition and in the implementation of the depression filling algorithm. In v2, a priority-flood + epsilon variant of the algorithm proposed by Barnes et al. (2014) is implemented. It prevents the formation of flat surfaces and allows for the determination of flow directions on all regions of the simulated landscape. The depression-less surface is then used to estimate depositional regions and to force marine deposition. An analyse of the differences between v1 and v2 on GitHub shows that there have been 36 commits over 12 files with 2,241 additions and 1,212 deletions. In addition, the first version published in JOSS (https://doi.org/10.21105/joss.00964) was a one-page summary that did not explain the details of the algorithms. This new publication describes in detail the physics and numerical approaches from eSCAPE, it also provides a series of hands-on examples that illustrate the code usage in different settings.

**GC2:** Furthermore, and I ask this out of naivety, how does eSCAPE differ from Bad- lands? Is the difference significant? Overall this is my only major concern, and it is one that is potentially wrong.

**Response:** There are many differences between Badlands and eSCAPE. First, the number of processes that can be simulated with eSCAPE is quite limited compared to Badlands. When considering the processes that both models simulate, the numerical approaches are completely different. Badlands is an explicit serial model able to simulate single flow direction river erosion/deposition. eSCAPE relies on an implicit iterative parallel approach able to evaluate multiple flow direction river processes. The approach in eSCAPE consists in solving a series of sparse matrix systems using the parallel library PETSc. In addition, eSCAPE can be used at global scale on a spherical mesh and relies on a different strategy to simulate depression filling (Planchon and Darboux 2001 for Badlands – Barnes 2014 for eSCAPE). In terms of outputs, one might find these two models similar, but they are really distinct when looking at the underlying algorithms and implementation strategies.

**Minor comments:**

**MC1:** The introduction way oversells the model. Yes, it can model global erosion and deposition using a set of rules, however, the model cannot capture lateral movement of the surface due to faulting. In fact, there is no faulting, which is arguably the major process that connects mantle convection to surface processes. This is a very challenging problem, and not one the author seeks to solve. However, much text is wasted on describing a vision of a global coupled model. This should be saved for a research proposal and not used here.

**Response:** Following reviewer's comment, I have modified the introduction and re- moved the paragraph related to the coupling with geodynamic/lithospheric models as this is not essential to the paper and I definitely do not want to oversell the model, as pointed out in the introduction: "The model presented in this paper is a first step toward the development of a parallel global scale landscape evolution model."

**MC2:** Explain what the advance is in this model, how it advances on v1 and Badlands. What is eSCAPE v2 for?

Response: See response to the general comments from the reviewer above (GC1 & GC2).

**MC3:** Line 15: What was the reason for cherry-picking these citations, none of which date from the '80s?

**Response:** Following the reviewer's comment, I have modified the text from the '80s to '90s. The choice of citations illustrates some of the LEM models that have been created over the years: Caesar (Coulthard), Cascade (Braun), Apero (Davy), Badlands (Salles) or Landlab (Hobley). In addition, these models represent different numerical approaches based on cellular automata, stream power law, or more standard flow hydrodynamics. They have also been developed to look at different spatial domains from river to catchment scale up to regional and continental extent.

**MC4:** Line 28: What is the purpose of this paragraph? As it is, it is far too short to encompass how global mantle flow is expressed at the earth's surface.

Response: Following the reviewer's comment, I have removed this paragraph from the introduction.

**MC5:** Line 70: I thought the approach of Jean Braun was O(N) efficient, always? Is the author saying otherwise?

**Response:** The approach from Braun is O(N) efficient but its parallel implementation relies on the number of outlets present in the simulation and therefore can become inefficient and scale poorly when the number of processors increases.

**MC6:** Equation 2: The first line does not make sense.  $q_1 = b_1$ , not  $q_i = b_i$

Response: I have made the correction in the manuscript

MC7: Line 127: "calibration" is out of place here.

Response: I have deleted "calibration"

**MC8:** Line 128: "evidence" should not get an "s", likewise "behaviour". There are other minor grammatical errors which I am sure will be corrected when copy edited.

Response: I have removed the "s" from evidence and behaviour in the text.

**MC9:** Equations 7 and 8: Here it is hard coded that n=1 and m=0.5. This is stated later in the manuscript, but this is potentially a major limitation of the model, as the recent study by Kwang & Parker (2017) suggests that "the choice m/n=0.5 yields a curiously unrealistic result: the predicted landscape is invariant to horizontal stretching".

**Response:** From Kwang & Parker (2017), this unrealistic behaviour is found when hillslope diffusion is neglected. In eSCAPE, hillslope diffusion could be turned on and thus should help to limit this behaviour. In addition, it is worth mentioning that the effect observed by Kwang & Parker is made when accounting only for a single flow direction (D8) when computing flow and drainage area. eSCAPE allows to simulate multiple flow direction (MDF) and the curious observations from Kwang & Parker have not been reported in such case.

**MC10:** Line 159: In this equation, the non-suspended sediment gets left behind, right? But the stream power law assumes instantaneous sediment transport. Therefore the two are incompatible? I am missing something here. Perhaps some additional explanation of how the model goes from erosion to deposition would help.

**Response:** At line 160, I define Ff as the fraction of fine sediment that remains in suspension. Ff represents the volumetric fraction of bedrock that breaks into sediment small enough to be considered permanently in suspension and for which no further treatment of bed–water column interactions is needed. For bedrock that breaks only into sand and gravel fractions, Ff would be zero. Therefore, simulated bed deposits and transported sediment flux only include sediment coarse enough that it does not permanently stay in suspension. I have added the explanation above in the manuscript.

**MC11:** Section 2.3: The "priority-flood" algorithm is non-physical, right? I wonder if it should not be done after the hillslope processes (diffusion), as this would smooth depressions and potentially fill them. Then the subsequent filling by fluvial deposition should occur?

**Response:** The reviewer is right, the "priority-flood" algorithm is a non-physical process and can be done prior to fluvial deposition. It could potentially help in cases where depressions are made of only a single point (local pit) or really small in size because induced filling from hillslope processes only occurs over much longer temporal scale than river ones. I believe over time, as the model iterates over the main loop the order proposed by the reviewer and the implemented one will produce equivalent results.

**MC12:** Section 2.5: Does marine deposition use a constant diffusion coefficient? Some marine deposition models vary this diffusion coefficient with water depth, to simulate wave and tide effects. I assume that this is not the case within eSCAPE?

**Response:** The reviewer is right I only use a constant diffusion coefficient for marine deposition in eSCAPE and do not account for water depth dependent (non-linear) dif- fusion. This could potentially be a new feature for the next model version.

MC13: Table 3: I think the marine parameters are missing from the table.

**Response:** The only user-defined parameter required to simulate marine processes is the diffusion parameter sedimentK defined in table 3 and at line 288 page 13.

**Reproducibility:**

**RC1:** The code is available, and I have successfully installed it. I have come across minor issues in running the code, due to my install of python and petsc, but this will be fixed before publication I am sure.

**Response:** I have made some changes in the code to fix some of the issues encountered by the reviewer (https://github.com/Geodels/eSCAPE/issues/9). I have also added some documentation about the petsc installation (https://github.com/Geodels/eSCAPE/wiki/Dependency)

**REVIEWER 2**

Again, I would like to thank the second reviewer for his useful comments. Below I provide a point by point response. I have also attached the updated manuscript based on the two reviews.

**Main comments:**

**MC1:** Although simulations at this scale are presented in the manuscript, the resolution (16 km) and the total number of mesh points (3 millions) used for these simulations remain somewhat coarse and small, respectively. The mesh size is not much greater than what could be processed nowadays at tractable computational costs using sequential model implementations. In this regard, it would be interesting (1) to see how eSCAPE performance does roughly compare with efficient, sequential implementation(s) of the same processes run on grids or meshes of a similar size (at least for the SFD / purely erosive case), and (2) to see how eSCAPE scales at much greater mesh sizes of, e.g., 10-100 millions of nodes (at least for the purely erosive case since applying pit resolving in serial might become computationally intractable at that scale). In my opinion, and this is my main concern, the manuscript could do a better job at showing when eSCAPE would become a compelling alternative to landscape evolution models based on sequential algorithms (graph traversal or other). The two suggestions above might help improving that.

**Response:** I will also be pleased to make such comparisons. Yet this is a pretty difficult task as (1) it is pretty hard to find authors that reports their computation time for their codes and (2) I have never used any code using an implicit approach like eSCAPE. From testing with the other code that I have access to (*e.g. Badlands*), eSCAPE for purely erosive simulations is performing much faster for mesh above half a million points thanks to the implicit approach that allows longer time step to be used even in serial. Yet the comparison is not that valuable as the algorithm design is very different. If one sets the model time steps to some smaller values, then Badlands will be more efficient... In other words, it is pretty hard to compare the efficiency of these codes if we set similar parameters. I found that the only potential comparison could be with *fastscape* code based on the Braun & Willett (2013) algorithm. In their paper they reported for a model similar to the one presented in section 3.2.1 (Fig. 3) that for a 100 million of nodes their code took 2.7 seconds per time step on eight cores. I did a test using the same parameters as the ones presented in section 3.2.1 and found the performance results presented in the Table below. I don't think that these results should be added to the manuscript as a proper benchmark as I haven't been using the same type of processors nor the exact same simulation parameters as the one in Braun & Willett (2013).

| Proc. # | Time step (s) |  |  |  |
|---------|---------------|--|--|--|
| 2       | 10.8          |  |  |  |
| 4       | 6.2           |  |  |  |
| 8       | 3.5           |  |  |  |
| 16      | 2.1           |  |  |  |

**MC2:** Besides that, I have a minor comment about the method chosen in eSCAPE to control the convergence vs. divergence of flow paths (SFD vs. MFD). Is there any specific reason why manually setting the number of flow receivers is preferred over unconditionally partitioning the outgoing flow between all downslope neighbors and let the user control the SFD vs MDF behavior, e.g., by tuning the parameter(s) of the weight vs. slope relationship? While both approaches to this problem are arbitrary, the second one would allow finer control and has been studied more in depth (see, e.g., Qin et al. 2007, https://doi.org/10.1080/13658810601073240). Or perhaps a combination of the two approaches would allow even greater flexibility.

**Response:** The algorithm actually uses an adaptive approach for determining the flow-partition between downslope neighbours as suggested by the reviewer. The description of this capability is found in the *MFDreceivers* function from the file 'fortran/functions.f90'. In my approach, the influence of local terrain on flow partition is modelled by a weight function which is based on local maximum downslope gradient. The manual setting of the number of flow receivers is there to improve the

efficiency of the approach. From testing, it appears that having a maximum number of downstream nodes set to 3 gives results similar to a full MFD approach.

**Specific comments:**

**MC1:** Line 69: Due to the issue (rightly pointed by the author in the following paragraph) of load balancing vs. the relative sizes of the catchments in the simulated domain, methods based on depth-first graph traversal may not scale at all in the worst case scenario of one single simulated catchment.

Response: I agree with the reviewer's comment!

**MC2:** Line 78: Note that approaches like the one described by Braun and Willet's (2013) can actually be easily modified to incorporate MFD algorithms. Unfortunately, no paper has been published yet on this.

**Response:** Thanks for the information, I haven't seen the application of Braun and Willet's algorithm to the MFD case but it will be a great addition for sure!

**MC3:** Fig 1a: I guess that the color map used for cell elevation values has been chosen so that it emphasizes dry (high) vs. wet (low) cells. Still, I doubt that the value of 4 meters has a special meaning, and a non-diverging color map would be more appropriate.

**Response:** The color scale represents elevation in (m) for a simple example used to illustrate flow paths on the triangular irregular network. Maybe the confusion comes from the fact that in addition to the cell color I have also written the nodes number (ID) on top of the figure. I have added this information to the figure caption.

MC4: Line 143: "m/y" units badly formatted.

**Response: Corrected**

**MC5:** Line 154: I might be missing something in the source code, but "scipy.sparse" is imported only in "surfprocplex.py" and it is not used further in that module. Maybe there is an inconsistency between the source code and the manuscript about how SciPy is used here?

**Response:** The reviewer is right *SciPy sparse matrix* is only called in the surfprocplex.py file and is required by petsc4py library to solve the system described in eq. (7). I have modified the text to specify that I use scipy sparse matrices (e.g. csr\_matrix) to efficiently load matrices into a petsc4py matrix.

Ref: https://bitbucket.org/petsc/petsc4py/issues/94/scatter-scipy-sparse-matrix-to-petsc-local

**MC6:** Line 178: It might be worth also mentioning the O(N) depression resolving algorithm (not part of the "priority-flood" family of algorithms) that has been published very recently by Cordonnier et al. (2019, https://doi.org/10.5194/esurf-7-549-2019). Disclaimer: I'm co-author of this paper.

**Response:** I have added the reference to Cordonnier et al. (2019) and this family of algorithms in the manuscript.

**MC7:** Section 2.4: It might be worth adding a few words on how the depression areas are delineated and how the spillover nodes are retrieved using the priority-flood + epsilon filling algorithm. This is not obvious, at least to me and potentially to other readers as well.

**Response:** To obtain the spillover nodes from the priority-flood + epsilon filling algorithm I use the approach proposed by Barnes (2017) in its parallel version (section 3.1 in the paper). I have added a reference to Barnes' work in the manuscript for this section: *"The spillover nodes are obtained using the method proposed by Barnes (2017) where in addition to depressions, the priority-flood approach labels watershed indices. Spillover nodes correspond to the lowest points connecting different*

watersheds. The updated elevation field is then used to compute the flow accumulation following the approach presented in section 2.1."

MC8: Line 380: Typo "be compare" -> "be compared".

**Response: Corrected**

**MC9:** Line 415: The Cordonnier et al.'s depression resolving algorithm cited here above is optimized specifically for use in landscape evolution models. Compared to the priority flood + epsilon variant of Barnes et al. (2014), it may drastically improve the overall performance when it is executed at every time step. That said, there is no parallel version yet and it works best when coupled with graph traversal algorithms.

**Response:** I have added the reference to the approach proposed by Cordonnier et al. (2019) in the corresponding section.

**eSCAPE: Regional to Global Scale Landscape Evolution Model v2.0**

Tristan Salles1

1School of Geosciences, University of Sydney, Sydney, NSW, 2006, Australia **Correspondence:** Tristan Salles (tristan.salles@sydney.edu.au)

**Abstract.** eSCAPE is a Python-based landscape evolution model that simulates over geological time (1) the dynamic of the landscape, (2) the transport of sediment from source to sink, and (3) continental and marine sedimentary basins formation under different climatic and tectonic conditions. eSCAPE is open-source, cross-platform, distributed under the GPLv3 license and available on GitHub (escape-model.github.io). Simulated processes rely on a simplified mathematical representation of

- 5 landscape processes the stream power and creep laws to compute Earth's surface evolution by rivers and hillslope transport. The main difference with previous models is in the underlying numerical formulation of the mathematical equations. The approach is based on a series of implicit iterative algorithms defined in matrix form to calculate both drainage area from multiple flow directions and erosion/deposition processes. eSCAPE relies on PETSc parallel library to solve these matrix systems. Along with the description of the algorithms, examples are provided and illustrate the model current capabilities and
- 10 limitations. eSCAPE is the first landscape evolution model able to simulate processes at global scale and is primarily designed to address problems on large unstructured grids (several millions of nodes).

Copyright statement. The article is distributed under the Creative Commons Attribution 4.0 License.

**1 Introduction**

[revised manuscript text omitted]

(8)

(7)

This system is implicit and the matrix is sparse. The SciPy compressed sparse row matrix functionality (Jones et al., 2001) is used to build  $\Gamma$  on local domains. The SciPy matrix format (*e.g.* csr\_matrix) is efficiently loaded as a PETSc Python matrix and the Eq. 7 is then solved using Richardson solver with block Jacobi preconditioning (*bjacobi*) using an initial guess for the

150

Once the entrainment rates have been obtained, the sediment flux moving out at every node  $Q_s^{out}$  equals the flux of sediment flowing in plus the local erosion rate.  $Q_s^{out}$  takes the following form:

$$Q_s^{out} = Q_s^{in} + (1 - F_f) E\Omega$$

solution set to vertices elevation.

155  $\Omega$  is the voronoi area of the considered vertex and  $F_f$  is the volumetric fraction of fine sediment small enough to be considered permanently in suspension. As an example, in case where bedrock breaks only into sand and gravel fractions,  $F_f$  would be zero. As a result, simulated deposits and transported sediment flux in the model only include sediment coarse enough that it does not permanently stay in suspension.

The solution of the above equation requires the calculation of the incoming sediment volume from upstream nodes  $Q_s^{in}$ . At node *i*, Eq. 8 is equivalent to:

$$q_{s,i} = e_i + \sum_{d=1}^{N_d} q_{s,d}$$
(9)

where  $e_i = (1 - F_f)E_i\Omega_i$  and  $N_d$  the number of donors. Assuming that river sediment concentration is distributed in a similar way as the water discharge we can write a similar set of equalities as the ones in Eq. 2. Then a matrix system as proposed for the FA (Eq. 4) can be obtained. The new system is then solved using the PETSc solver and preconditioner previously defined.

**165 2.3 Priority-flood depression filling**

In most landscape evolution models, internally-draining regions (e.g., depressions and pits) are usually filled before the calculation of flow discharge and erosion-deposition rates. This ensures that all flows conveniently reach the coast or the boundary of the simulated domain. In models intended to simulate purely erosional features, such depressions are usually treated as transient features and often ignored. However, eSCAPE is designed to not only address erosion problems but also to simulate

- 170 source-to-sink transfer and sedimentary basins formation and evolution in potentially complex tectonic settings. In such cases, depressions may be formed at different periods during runtime and may be filled or remain internally drained (e.g., endorheic basins) depending on the volume of sediment transported by upstream catchments.
- Depression filling approaches have received some attention in recent years with the development of new and more efficient algorithms (Wang and Liu, 2006; Barnes et al., 2014; Zhou et al., 2016, 2017; Wei et al., 2018). These methods based on
  priority-flood offer a time complexity of the order of O(Nlog(N)) compared to older approaches such as the Jenson and Domingue (1988) (O(N2)) or Planchon and Darboux (2002) (O(N1.2)) algorithms.

Priority-flood algorithms consist in finding the minimum elevation a cell needs to be raised to (e.g., spill elevation of a cell) to prevent downstream ascending path to occur. They rely on priority queue data structure used to efficiently find the lowest spill elevation in a grid. Depending on the chosen method, priority queue implementation approaches affect the time complexity

- 180 of the algorithm (Barnes et al., 2014). In eSCAPE, the priority-flood +  $\epsilon$  variant of the algorithm proposed in Barnes et al. (2014) is implemented. It provides a solution to remove automatically flat surfaces and it produces surfaces for which each cell has a defined gradient from which flow directions can be determined. Recently, Cordonnier et al. (2019) proposed a different potentially more efficient approach based on a O(N) depression resolving algorithm that explicit compute the flow paths through the construction of a graph connecting together all adjacent drainage basins.
- 185 In eSCAPE, this part of the algorithm is not parallelised and is performed on the master processor. It starts from the grid border vertices and processes vertices that are in their immediate neighbourhoods one by one in the ascending order of their spill elevations (Barnes et al., 2014). The initialisation step consists in pushing all the edges nodes onto a priority queue. The priority queue rearranges these nodes so that the ones with the lowest elevations in the queue are always processed first.

---

## Author Response (AR1)

**REVIEWER 1**

First, I would like to thank the reviewer for his useful comments. Below is my response to these comments.

**General comments:**

**GC1:** *I would like to hear how the author defends criticism that this model is not novel. eSCAPE v1 was published in the Journal of Open Source Software, how does v2 differ? What makes it require a whole new publication?*

**Response:** The main differences between v1 and v2 of eSCAPE are in the way v2 handles the marine deposition and in the implementation of the depression filling algorithm. In v2, a priority-flood + epsilon variant of the algorithm proposed by Barnes et al. (2014) is implemented. It prevents the formation of flat surfaces and allows for the determination of flow directions on all regions of the simulated landscape. The depression-less surface is then used to estimate depositional regions and to force marine deposition. An analyse of the differences between v1 and v2 on GitHub shows that there have been 36 commits over 12 files with 2,241 additions and 1,212 deletions. In addition, the first version published in JOSS ([https://doi.org/10.21105/joss.00964](https://doi.org/10.21105/joss.00964)) was a one-page summary that did not explain the details of the algorithms. This new publication describes in detail the physics and numerical approaches from eSCAPE, it also provides a series of hands-on examples that illustrate the code usage in different settings.

**GC2:** *Furthermore, and I ask this out of naivety, how does eSCAPE differ from Bad- lands? Is the difference significant? Overall this is my only major concern, and it is one that is potentially wrong.*

**Response:** There are many differences between Badlands and eSCAPE. First, the number of processes that can be simulated with eSCAPE is quite limited compared to Badlands. When considering the processes that both models simulate, the numerical approaches are completely different. Badlands is an explicit serial model able to simulate single flow direction river erosion/deposition. eSCAPE relies on an implicit iterative parallel approach able to evaluate multiple flow direction river processes. The approach in eSCAPE consists in solving a series of sparse matrix systems using the parallel library PETSc. In addition, eSCAPE can be used at global scale on a spherical mesh and relies on a different strategy to simulate depression filling (Planchon and Darboux 2001 for Badlands – Barnes 2014 for eSCAPE). In terms of outputs, one might find these two models similar, but they are really distinct when looking at the underlying algorithms and implementation strategies.

**Minor comments:**

**MC1:** T*he introduction way oversells the model. Yes, it can model global erosion and deposition using a set of rules, however, the model cannot capture lateral movement of the surface due to faulting. In fact, there is no faulting, which is arguably the major process that connects mantle convection to surface processes. This is a very challenging problem, and not one the author seeks to solve. However, much text is wasted on describing a vision of a global coupled model. This should be saved for a research proposal and not used here.*

**Response:** Following reviewer's comment, I have modified the introduction and re- moved the paragraph related to the coupling with geodynamic/lithospheric models as this is not essential to the paper and I definitely do not want to oversell the model, as pointed out in the introduction: "The model presented in this paper is a first step toward the development of a parallel global scale landscape evolution model."

**MC2:** *Explain what the advance is in this model, how it advances on v1 and Badlands. What is eSCAPE v2 for?*

**Response:** See response to the general comments from the reviewer above (GC1 & GC2).

**MC3:** *Line 15: What was the reason for cherry-picking these citations, none of which date from the '80s?*

**Response:** Following the reviewer's comment, I have modified the text from the '80s to '90s. The choice of citations illustrates some of the LEM models that have been created over the years: Caesar (Coulthard), Cascade (Braun), Apero (Davy), Badlands (Salles) or Landlab (Hobley). In addition, these models represent different numerical approaches based on cellular automata, stream power law, or more standard flow hydrodynamics. They have also been developed to look at different spatial domains from river to catchment scale up to regional and continental extent.

**MC4:** *Line 28: What is the purpose of this paragraph? As it is, it is far too short to encompass how global mantle flow is expressed at the earth's surface.*

**Response:** Following the reviewer's comment, I have removed this paragraph from the introduction.

**MC5:** *Line 70: I thought the approach of Jean Braun was O(N) efficient, always? Is the author saying otherwise?*

**Response:** The approach from Braun is O(N) efficient but its parallel implementation relies on the number of outlets present in the simulation and therefore can become inefficient and scale poorly when the number of processors increases.

**MC6:** *Equation 2: The first line does not make sense. $q\_1 = b\_1$, not $q\_i = b\_i$*

**Response:** I have made the correction in the manuscript

**MC7:** *Line 127: "calibration" is out of place here.*

**Response:** I have deleted "calibration"

**MC8:** *Line 128: "evidence" should not get an "s", likewise "behaviour". There are other minor grammatical errors which I am sure will be corrected when copy edited.*

**Response:** I have removed the "s" from evidence and behaviour in the text.

**MC9:** *Equations 7 and 8: Here it is hard coded that n=1 and m=0.5. This is stated later in the manuscript, but this is potentially a major limitation of the model, as the recent study by Kwang & Parker (2017) suggests that "the choice m/n=0.5 yields a curiously unrealistic result: the predicted landscape is invariant to horizontal stretching".*

**Response:** From Kwang & Parker (2017), this unrealistic behaviour is found when hillslope diffusion is neglected. In eSCAPE, hillslope diffusion could be turned on and thus should help to limit this behaviour. In addition, it is worth mentioning that the effect observed by Kwang & Parker is made when accounting only for a single flow direction (D8) when computing flow and drainage area. eSCAPE allows to simulate multiple flow direction (MDF) and the curious observations from Kwang & Parker have not been reported in such case.

**MC10***: Line 159: In this equation, the non-suspended sediment gets left behind, right? But the stream power law assumes instantaneous sediment transport. Therefore the two are incompatible? I am missing something here. Perhaps some additional explanation of how the model goes from erosion to deposition would help.*

**Response:** At line 160, I define Ff as the fraction of fine sediment that remains in suspension. Ff represents the volumetric fraction of bedrock that breaks into sediment small enough to be considered permanently in suspension and for which no further treatment of bed–water column interactions is needed. For bedrock that breaks only into sand and gravel fractions, Ff would be zero. Therefore, simulated bed deposits and transported sediment flux only include sediment coarse enough that it does not permanently stay in suspension. I have added the explanation above in the manuscript.

**MC11:** *Section 2.3: The "priority-flood" algorithm is non-physical, right? I wonder if it should not be done after the hillslope processes (diffusion), as this would smooth depressions and potentially fill them. Then the subsequent filling by fluvial deposition should occur?*

**Response:** The reviewer is right, the "priority-flood" algorithm is a non-physical process and can be done prior to fluvial deposition. It could potentially help in cases where depressions are made of only a single point (local pit) or really small in size because induced filling from hillslope processes only occurs over much longer temporal scale than river ones. I believe over time, as the model iterates over the main loop the order proposed by the reviewer and the implemented one will produce equivalent results.

**MC12:** *Section 2.5: Does marine deposition use a constant diffusion coefficient? Some marine deposition models vary this diffusion coefficient with water depth, to simulate wave and tide effects. I assume that this is not the case within eSCAPE?*

**Response:** The reviewer is right I only use a constant diffusion coefficient for marine deposition in eSCAPE and do not account for water depth dependent (non-linear) dif- fusion. This could potentially be a new feature for the next model version.

**MC13:** *Table 3: I think the marine parameters are missing from the table.*

**Response:** The only user-defined parameter required to simulate marine processes is the diffusion parameter sedimentK defined in table 3 and at line 288 page 13.

**Reproducibility:**

**RC1:** *The code is available, and I have successfully installed it. I have come across minor issues in running the code, due to my install of python and petsc, but this will be fixed before publication I am sure.*

**Response:** I have made some changes in the code to fix some of the issues encountered by the reviewer (https://github.com/Geodels/eSCAPE/issues/9). I have also added some documentation about the petsc installation (https://github.com/Geodels/eSCAPE/wiki/Dependency)

**REVIEWER 2**

Again, I would like to thank the second reviewer for his useful comments. Below I provide a point by point response. I have also attached the updated manuscript based on the two reviews.

**Main comments:**

**MC1:** *Although simulations at this scale are presented in the manuscript, the resolution (16 km) and the total number of mesh points (3 millions) used for these simulations remain somewhat coarse and small, respectively. The mesh size is not much greater than what could be processed nowadays at tractable computational costs using sequential model implementations. In this regard, it would be interesting (1) to see how eSCAPE performance does roughly compare with efficient, sequential implementation(s) of the same processes run on grids or meshes of a similar size (at least for the SFD / purely erosive case), and (2) to see how eSCAPE scales at much greater mesh sizes of, e.g., 10-100 millions of nodes (at least for the purely erosive case since applying pit resolving in serial might become computationally intractable at that scale). In my opinion, and this is my main concern, the manuscript could do a better job at showing when eSCAPE would become a compelling alternative to landscape evolution models based on sequential algorithms (graph traversal or other). The two suggestions above might help improving that.*

**Response:** I will also be pleased to make such comparisons. Yet this is a pretty difficult task as (1) it is pretty hard to find authors that reports their computation time for their codes and (2) I have never used any code using an implicit approach like eSCAPE. From testing with the other code that I have access to (*e.g. Badlands*), eSCAPE for purely erosive simulations is performing much faster for mesh above half a million points thanks to the implicit approach that allows longer time step to be used even in serial. Yet the comparison is not that valuable as the algorithm design is very different. If one sets the model time steps to some smaller values, then Badlands will be more efficient… In other words, it is pretty hard to compare the efficiency of these codes if we set similar parameters. I found that the only potential comparison could be with *fastscape* code based on the Braun & Willett (2013) algorithm. In their paper they reported for a model similar to the one presented in section 3.2.1 (Fig. 3) that for a 100 million of nodes their code took 2.7 seconds per time step on eight cores. I did a test using the same parameters as the ones presented in section 3.2.1 and found the performance results presented in the Table below. I don't think that these results should be added to the manuscript as a proper benchmark as I haven't been using the same type of processors nor the exact same simulation parameters as the one in Braun & Willett (2013).

| Proc. # | Time step (s) |
|---------|---------------|
| 2 | 10.8 |
| 4 | 6.2 |
| 8 | 3.5 |
| 16 | 2.1 |

**MC2:** *Besides that, I have a minor comment about the method chosen in eSCAPE to control the convergence vs. divergence of flow paths (SFD vs. MFD). Is there any specific reason why manually setting the number of flow receivers is preferred over unconditionally partitioning the outgoing flow between all downslope neighbors and let the user control the SFD vs MDF behavior, e.g., by tuning the parameter(s) of the weight vs. slope relationship? While both approaches to this problem are arbitrary, the second one would allow finer control and has been studied more in depth (see, e.g., Qin et al. 2007, https://doi.org/10.1080/13658810601073240). Or perhaps a combination of the two approaches would allow even greater flexibility.*

**Response:** The algorithm actually uses an adaptive approach for determining the flow-partition between downslope neighbours as suggested by the reviewer. The description of this capability is found in the *MFDreceivers* function from the file 'fortran/functions.f90'. In my approach, the influence of local terrain on flow partition is modelled by a weight function which is based on local maximum downslope gradient. The manual setting of the number of flow receivers is there to improve the

efficiency of the approach. From testing, it appears that having a maximum number of downstream nodes set to 3 gives results similar to a full MFD approach.

**Specific comments:**

**MC1:** *Line 69: Due to the issue (rightly pointed by the author in the following paragraph) of load balancing vs. the relative sizes of the catchments in the simulated domain, methods based on depth-first graph traversal may not scale at all in the worst case scenario of one single simulated catchment.*

**Response:** I agree with the reviewer's comment!

**MC2:** *Line 78: Note that approaches like the one described by Braun and Willet's (2013) can actually be easily modified to incorporate MFD algorithms. Unfortunately, no paper has been published yet on this.*

**Response:** Thanks for the information, I haven't seen the application of Braun and Willet's algorithm to the MFD case but it will be a great addition for sure!

**MC3:** *Fig 1a: I guess that the color map used for cell elevation values has been chosen so that it emphasizes dry (high) vs. wet (low) cells. Still, I doubt that the value of 4 meters has a special meaning, and a non-diverging color map would be more appropriate.*

**Response:** The color scale represents elevation in (m) for a simple example used to illustrate flow paths on the triangular irregular network. Maybe the confusion comes from the fact that in addition to the cell color I have also written the nodes number (ID) on top of the figure. I have added this information to the figure caption.

**MC4:** *Line 143: "m/y" units badly formatted.*

**Response:** Corrected

**MC5:** *Line 154: I might be missing something in the source code, but "scipy.sparse" is imported only in "surfprocplex.py" and it is not used further in that module. Maybe there is an inconsistency between the source code and the manuscript about how SciPy is used here?*

**Response:** The reviewer is right *SciPy sparse matrix* is only called in the surfprocplex.py file and is required by petsc4py library to solve the system described in eq. (7).  I have modified the text to specify that I use scipy sparse matrices (e.g. csr_matrix) to efficiently load matrices into a petsc4py matrix.
Ref: https://bitbucket.org/petsc/petsc4py/issues/94/scatter-scipy-sparse-matrix-to-petsc-local

**MC6:** *Line 178: It might be worth also mentioning the O(N) depression resolving algorithm (not part of the "priority-flood" family of algorithms) that has been published very recently by Cordonnier et al. (2019, https://doi.org/10.5194/esurf-7-549-2019). Disclaimer: I'm co-author of this paper.*

**Response:** I have added the reference to Cordonnier et al. (2019) and this family of algorithms in the manuscript.

**MC7:** *Section 2.4: It might be worth adding a few words on how the depression areas are delineated and how the spillover nodes are retrieved using the priority-flood + epsilon filling algorithm. This is not obvious, at least to me and potentially to other readers as well.*

**Response:** To obtain the spillover nodes from the priority-flood + epsilon filling algorithm I use the approach proposed by Barnes (2017) in its parallel version (section 3.1 in the paper). I have added a reference to Barnes' work in the manuscript for this section: *"The spillover nodes are obtained using the method proposed by Barnes (2017) where in addition to depressions, the priority-flood approach labels watershed indices. Spillover nodes correspond to the lowest points connecting different*

*watersheds. The updated elevation field is then used to compute the flow accumulation following the approach presented in section 2.1."*

**MC8:** *Line 380: Typo "be compare" -> "be compared".*

**Response:** Corrected

**MC9:** *Line 415: The Cordonnier et al.'s depression resolving algorithm cited here above is optimized specifically for use in landscape evolution models. Compared to the priority flood + epsilon variant of Barnes et al. (2014), it may drastically improve the overall performance when it is executed at every time step. That said, there is no parallel version yet and it works best when coupled with graph traversal algorithms.*

**Response:** I have added the reference to the approach proposed by Cordonnier et al. (2019) in the corresponding section.

**eSCAPE: Regional to Global Scale Landscape Evolution Model v2.0**

Tristan Salles[1]

[1]School of Geosciences, University of Sydney, Sydney, NSW, 2006, Australia

**Correspondence:** Tristan Salles (tristan.salles@sydney.edu.au)

**Abstract.**  eSCAPE is a Python-based landscape evolution  model that simulates over geological time (1)  the dynamic of the landscape, (2)  the transport of sediment from source to sink, and (3) continental and marine sedimentary basins formation under different climatic and tectonic conditions. eSCAPE is open-source, cross-platform, distributed under the GPLv3 license and available on GitHub (escape-model.github.io). Simulated processes  rely on a simplified mathematical representation of landscape processes - the stream power and creep laws - to compute Earth's surface evolution by rivers and hillslope transport. The main difference with previous models is in the underlying numerical formulation of the mathematical equations. The approach is based on a series of implicit iterative algorithms defined in matrix form to calculate both drainage area from multiple flow directions and erosion/deposition processes. eSCAPE relies on PETSc parallel library to solve these matrix systems. Along with the  description of the algorithms, examples are provided and illustrate the model current capabilities and limitations. eSCAPE is the first landscape evolution model able to simulate processes at global scale and is primarily designed to address problems on large unstructured grids (several millions of nodes).

*Copyright statement.* The article is distributed under the Creative Commons Attribution 4.0 License.

**1 Introduction**

 Since the '90s, many software have been designed to estimate long-term catchment dynamic, drainage evolution as well as sedimentary basins formation in response to various mechanisms such as tectonic or climatic forcing (Braun and Sambridge, 1997; Coulthard et al., 2002; Davy and Lague, 2009; Simoes et al., 2010; Salles, 2016; Grieve et al., 2016b; Hobley et al., 2017). These models  rely on a set of mathematical  and physical expressions that simulate sediment erosion, transport and deposition and can reproduce the first order complexity of Earth's surface geomorphological evolution (Tucker and Hancock, 2010; Shobe et al., 2017).

In most of these models, climatic and tectonic conditions are imposed and often consist in rather simple forcing such as uniform
spatial precipitation and vertical displacements (uplift or subsidence) far from reflecting the complexity of the natural system. In
addition such approaches are unable to properly explore potential feedback mechanisms between each of the Earth components.
In fact, only a handful of these models are able to account more completely for the dynamics of the lithosphere and mantle, the
role of sedimentation and provide a more quantitative representation of climate relative to its interactions with topography (such
as orographic rain)
(Beaumont et al., 1992; Salles et al., 2011; Bianchi et al., 2015; Thieulot et al., 2014; Yang et al., 2015; Salles et al., 2017; Beucher et al.,
. When made possible, it is often realised through the coupling of specialised numerical models involving the expertise of geo-
dynamicists, geophysicists, Earth surface and atmospheric scientists.

~~Many advanced numerical models of tectonic processes constrained by geological and geophysical observations have been developed and global scale geodynamic models (Zhong et al., 2000; Moresi et al., 2003; Heister et al., 2017) have shown how mantle convection drives the motion of tectonic plates and dictates the long-term evolution of the Earth. Similarly progresses in the understanding of past, present, and future climates have been made by the development of mathematical models of the general circulation of a planetary atmosphere or ocean that simulate climate at an increasing level of detail (Dutkiewicz et al., 2016; Brown et al., 2018).~~

Yet, we are still missing a tool to evaluate global scale evolution of Earth surface and its interaction with the atmosphere, the
hydrosphere, the tectonic and mantle dynamics. Such a tool will certainly provide new insights and help to better characterise
many aspects of the Earth system ranging from the role of atmospheric circulation on physical denudation, from the influence
of erosion and deposition of sediments on mantle convection, from the location and abundance of natural resources to the
evolution of life.

The model presented in this paper is a first step toward the development of a parallel global scale landscape evolution model. It
 allows to couple the Earth's surface with global climatic perturbations and geodynamic
forces acting within the Earth's interior. Landscapes and sedimentary basins evolution in eSCAPE are driven by a series of
standard stream power incision and diffusion laws (Howard et al., 1994; Tucker and Slingerland, 1997; Chen et al., 2014)
designed to address problems from regional to global scales and over geological time ($10^5$-$10^9$ years). Due to the inherent
assumptions made in the set of equations used, eSCAPE is not intended to estimate the evolution of individual fluvial channels
but to quantify large scale and long term evolution of Earth's surface regions (Salles et al., 2017; Armitage, 2019). ~~It is worth mentioning that eSCAPE simulates sediment supply and routing from source to sink in a self-consistent manner. In other words, the erosion occurring in upstream catchments is linked to sedimentation on basin margins through sediment routing resulting from a combination of channelling and hillslope processes. Sediment supply to continental margins is dynamically determined and results from both allogenic causes (*e.g.* the interactions with tectonic and/or climatic forcing or eustatic variations) and autogenic changes like the ones induced on catchments physiography.~~

[revised manuscript text omitted]

Once the entrainment rates have been obtained, the sediment flux moving out at every node $Q_s^{out}$ equals the flux of sediment

170   flowing in plus the local erosion rate. $Q_s^{out}$ takes the following form:

$$Q_s^{out} = Q_s^{in} + (1 - F_f) E \Omega$$

$\Omega$ is the voronoi area of the considered vertex and $F_f$ is the volumetric fraction of fine sediment that remains small enough to be considered permanently in suspension. As an example, in case where bedrock breaks only into sand and gravel fractions, $F_f$

would be zero. As a result, simulated deposits and transported sediment flux in the model only include sediment coarse enough that it does not permanently stay in suspension.

[revised manuscript text omitted]